# A path integral ground state Monte Carlo algorithm for entanglement of lattice bosons

Emanuel Casiano-Diaz[1,2], C. M. Herdman[3] and Adrian Del Maestro[1,4,5*]

**1** Department of Physics and Astronomy, University of Tennessee, Knoxville, TN 37996, USA
**2** Los Alamos National Laboratory, Computer, Computational and Statistical Sciences Division, Los Alamos, NM 87545, USA
**3** Department of Physics, Middlebury College, Middlebury, VT 05753, USA
**4** Min H. Kao Department of Electrical Engineering and Computer Science, University of Tennessee, Knoxville, TN 37996, USA
**5** Institute for Advanced Materials and Manufacturing, University of Tennessee, Knoxville, TN 37996, USA

★ Adrian.DelMaestro@utk.edu

## Abstract

A ground state path integral quantum Monte Carlo algorithm is introduced that allows for the study of entanglement in lattice bosons at zero temperature. The Rényi entanglement entropy between spatial subregions is explored across the phase diagram of the one dimensional Bose-Hubbard model for systems consisting of up to $L = 256$ sites at unit-filling without any restrictions on site occupancy, far beyond the reach of exact diagonalization. The favorable scaling of the algorithm is demonstrated through a further measurement of the Rényi entanglement entropy at the two dimensional superfluid-insulator critical point for large system sizes, confirming the existence of the expected entanglement boundary law in the ground state. The Rényi estimator is extended to measure the symmetry resolved entanglement that is operationally accessible as a resource for experimentally relevant lattice gases with fixed total particle number.


# 1  Introduction

The advent of replica based methods in quantum many-body systems have provided a route to measuring entanglement entropies – a quantification of the amount of non-classical information shared between a bipartition of a pure state – without the need to construct the full

density matrix via full state tomography [1]. Instead, the purity (related to the Rényi entanglement entropy) can be directly obtained via the expectation value of a local observable that is accessible in experimental quantum simulators based on ultra-cold lattice gases [2]. While the replica trick and its related SWAP algorithm have been implemented in numerical quantum Monte Carlo simulations to measure Rényi entropies at zero temperature [3–7] and the associated mutual information for finite temperature mixed states [8–11] in many physical scenarios (e.g. localized spin systems, lattice fermions, and continuum quantum fluids), they have yet to be extended to the case of fully itinerant and indistinguishable softcore lattice bosons where experiments are presently possible [12, 13].

The bosonic Hamiltonians of relevance to these systems, both in the continuum and on a lattice, can be exactly simulated without a sign problem in any dimension via path integral Monte Carlo (PIMC) [14–16]. Of particular importance is the Worm Algorithm [14, 17–20], which expands the configuration space of $D + 1$ dimensional worldlines to include discontinuous paths representing finite particle trajectories in imaginary time. The imaginary-time dynamics of these *worms* improve ergodicity and allow for the direct sampling of the bosonic Hilbert space at finite temperature [17, 21], and open source packages implementing the algorithm are available [22, 23]. At zero temperature, PIMC has a projector variant known as path integral ground state Monte Carlo (PIGS) that has been previously implemented for non-relativistic bosons in the spatial continuum [24, 25], with other Monte Carlo methods inspired by the PIGS formalism applied to spin models and fermionic lattices [26]. A continuous imaginary time $T = 0$ variant of PIMC for lattice bosons has been elusive in the literature. This algorithmic gap is relevant to the physical modeling of experiments of ultracold atoms confined to optical lattices [27], where finite temperature PIMC require an extrapolation in temperature to properly describe ground state properties [28, 29], as well as the measurement of entanglement [12, 13] highlighted above.

In this paper, we address both of these issues by introducing a zero temperature worm algorithm projector lattice quantum Monte Carlo algorithm (PIGS For Lattice Implementations or `PIGSFLI`) extending finite temperature PIMC to ground state calculations [30] where the Rényi entanglement entropy can be computed. Its domain of applicability includes any $D$-dimensional bosonic lattice Hamiltonian with arbitrary range interactions and hopping and it scales linearly in the total number of lattice sites ($L^D$) and the projection length $\beta$ such that Monte Carlo updates in the algorithm scale as O($\beta L^D$). The quantum Monte Carlo (QMC) method operates in the Fock space of bosonic occupation vectors $|n_1, \ldots, n_{L^D}\rangle$ with projection to the ground state proceeding from a trial state $|\Psi_T\rangle$. Here, a physically motivated choice for $|\Psi_T\rangle$ can lead to a significant acceleration in algorithm convergence. In practice, any ground state expectation value can be obtained to arbitrary precision by performing simulations at different values of the imaginary time projection length $\beta$ and performing an exponential fit to extract the $\beta \to \infty$ result.

Working within an expanded configuration space consisting of $\alpha$ independent copies of the imaginary time worldlines, the *replica trick* [1] is exploited to derive an efficient estimator for the $\alpha^{\text{th}}$ Rényi entanglement entropy using the SWAP algorithm [3] adapted to bosonic Hilbert space. While the `PIGSFLI` algorithm naturally operates in the grand canonical ensemble with the average filling fraction $\langle n \rangle = \langle N \rangle / L^D$ controlled by a chemical potential $\mu$, by restricting updates that change the number of particles away from a target value ($N$), the canonical ensemble can also be efficiently simulated at $T = 0$. In this case, where the number of particles is fixed, the symmetry resolved Rényi entanglement entropy [31–33] can be computed by projecting into the subspace of fixed local particle number in a spatial subregion. This latter quantity is important as it sets an upper bound on the amount of entanglement that could be extracted from the many-body system and transferred to a qubit register via local operations and classical communication (LOCC) [34].

The algorithm and proposed estimators are carefully benchmarked against exact diagonalization results for the Bose-Hubbard model. We find that relative errors of order $10^{-4}$ can be obtained for both the kinetic and potential energies and for unconventional estimators like the Rényi entanglement entropies (both full and symmetry resolved), errors as small as $10^{-3}$. Extrapolation to $\beta \to \infty$ can be performed using only a few finite $\beta$ simulations with the largest value required for a bias-free fit scaling with system size $L$. While in one spatial dimension, the density matrix renormalization group (DMRG) can be used to obtain the spatial entanglement entropy in this model [35], recent work suggests that the required restriction of the local Hilbert space to a fixed number of bosons can lead to errors in the symmetry resolved entanglement that grows logarithmically in the system size for weak, but finite soft-core repulsion [36]. In two spatial dimensions, for lattice sizes up to 1024 sites at the critical point between a superfluid and insulator, we demonstrate a perimeter law, producing high quality data that is suitable for the extraction of logarithmic corrections which can provide information on the underlying gapless excitations in the system.

A summary of the most important contributions of this paper include: (1) the introduction of the ground state `PIGSFLI` algorithm and associated open source code base [30] and new estimators for the efficient measurement of Rényi entanglement entropies within the lattice path integral framework. (2) Results for the spatial entanglement entropy in the 1D and 2D Bose-Hubbard model both at the quantum critical point, and across the superfluid-insulator phase diagram for much larger system sizes than had been previously studied, and without any local restrictions on bosonic site occupations. (3) A finite size scaling analysis of subleading corrections near the 2D superfluid-insulator quantum critical point which can encode universal properties of the interacting system. (4) Particle number distributions and symmetry resolved entanglement in the superfluid, critical, and Mott insulating regimes of the 1D Bose-Hubbard model. A strong dependence of the symmetry resolved entanglement with respect to the local particle number is interpreted in terms of interactions between quasiparticles. We believe the new algorithm has wide utility in the measurement and quantification of quantum correlations in bosonic lattice models and can be extended to model current and next-generation experiments on lattice gases. The ability to measure the symmetry resolved (and operationally accessible) entanglement in such systems has direct implications for the exploitation of correlated superfluid and insulating phases as entanglement resources for quantum information processing.

In the remainder of this paper, we introduce the theoretical concepts underlying the Rényi entanglement entropy, before providing an introduction to path integral Monte Carlo on a lattice, with emphasis on the algorithmic extensions required for it to operate in a projector form at zero temperature. The new replicated bosonic configuration space and Monte Carlo updates necessary for the measurement of entanglement entropy are described before benchmarking on small system sizes that are amenable to exact diagonalization. We conclude with a discussion of natural applications of the method to explore the scaling of symmetry resolved and operationally accessible entanglement in higher dimensions.

To facilitate the reproduction of results presented in this work, and to promote further exploration using the `PIGSFLI` algorithm, the `C++` source code has been released [30]. The scripts used to process the raw Monte Carlo data [37], along with the processed data files and scripts used to generate all plots in this paper are also available in a public repository [38].

## 2 Entanglement Entropy

Entanglement quantifies the non-classical correlations present in a joint state of a quantum system. Its characterization requires defining a partition of the system into subsystems; here

we only consider a bipartition into a spatial subregion $A$ and its complement $B$, however other types of bipartitions are also interesting, including in terms of particles (see e.g. [6,36,39,40]). Given a pure state $|\Psi\rangle$, the reduced density matrix of the $A$ subsystem is defined to be:

$$\rho_A = \text{Tr}_B |\Psi\rangle\langle\Psi| . \tag{1}$$

In general, $\rho_A$ describes a mixed state due to entanglement between $A$ and $B$ which can be quantified by the Rényi entanglement entropy (EE):

$$S_\alpha(\rho_A) = \frac{1}{1-\alpha} \ln \text{Tr}\, \rho_A^\alpha . \tag{2}$$

For $\alpha \to 1$, the Rényi entanglement entropy reduces to the von Neumann entanglement entropy: $S_1(\rho_A) = -\text{Tr}\,\rho_A \ln \rho_A$. Despite the fact that $S_\alpha$ as defined in Eq. (2) is not in the form of an expectation value of an observable, computational methods have been developed to compute Eq. (2) for many-body systems in Monte Carlo simulations [3]. Moreover, certain experimental many-body systems have the capability to directly experimentally measure Eq. (2) [28,29].

The entanglement entropy has been studied in a wide array of quantum many-body systems [41], providing important insights into the nature of quantum correlations. In particular, for the ground states of interacting many-body systems, the scaling of the entanglement entropy with subsystem size can display universal features of phases of matter [42]. Generically, ground states display an "area-law" scaling, where the entanglement entropy grows with the size of the boundary between subregions [43–45]. For systems with gapless excitations, additional terms appear in the entanglement entropy that either scale logarithmically with the boundary, or are independent of boundary size; the dimensionless coefficients of these terms characterize universal features of such phases of matter, such as the number of Goldstone modes [46–48] or the central charge of the underlying conformal field theory [49].

## 2.1 Symmetry-resolved and accessible entanglement entropies

In physical systems which conserve particle number (such as trapped ultracold gases), the amount of entanglement that is operationally accessible using local operations and classical communications (LOCC) is limited by the superselection rule that forbids creating superpositions of different particle number [31]. For $\alpha = 1$, the von Neumann accessible entanglement entropy is simply a weighted average of the entanglement entropies for $\rho_A$ projected onto a fixed subsystem particle number, known as the symmetry-resolved entanglement entropies $S_1(\rho_{A_n})$ [32]:

$$S_1^{\text{acc}}(\rho_A) = \sum_n P_n S_1(\rho_{A_n}) . \tag{3}$$

In Eq. (3) $n$ is the number of particles in subregion $A$, $P_n$ is the probability of $A$ having $n$ particles $P_n \equiv \text{Tr}(\Pi_n \rho_A \Pi_n)$, where $\Pi_n$ is a projector onto the subspace of $A$ with $n$ particles, and $\rho_{A_n}$ is the reduced density matrix of $A$, projected onto fixed local particle number $n$:

$$\rho_{A_n} = \frac{1}{P_n} \Pi_n \rho_A \Pi_n . \tag{4}$$

For the Rényi entropy [33] for general $\alpha$, the operationally accessible entanglement is

$$S_\alpha^{\text{acc}}(\rho_A) = \frac{\alpha}{1-\alpha} \ln \left[ \sum_n P_n e^{\frac{1-\alpha}{\alpha} S_\alpha(\rho_{A_n})} \right], \tag{5}$$

which reduces to $S_1^{\text{acc}}$ for $\alpha \to 1$.

$S_\alpha^{\text{acc}}$ represents an experimentally relevant bound on the entanglement that may be extracted from systems of indistinguishable and itinerant non-relativistic particles. The quantification of the symmetry-resolved and accessible entanglement entropies has garnered much interest recently in systems of both non-interacting and interacting particles [32–34, 36, 50–84]. We show in Section 4.3 that both the symmetry resolved entanglement entropy, $S_\alpha(\rho_{A_n})$, and the accessible entanglement entropy, $S_\alpha^{\text{acc}}$, can be computed for interacting boson systems using Monte Carlo methods, opening up a number of exciting potential avenues of study.

# 3 Path Integral Ground State Quantum Monte Carlo

Path Integral Ground State (PIGS) quantum Monte Carlo utilizes the projection of a trial wave function in imaginary time to obtain stochastically exact results for the ground state of a quantum many-body system. It has been previously formulated in first quantization for non-relativistic Hamiltonians in the spatial continuum [16, 24] and in second quantization on a lattice at finite temperature [15, 17, 22, 23]. Here we present the $T = 0$ projector formalism for lattice systems, focusing on the imaginary time wordlines of a local bosonic Hamiltonian.

## 3.1 Projection onto the Ground State

The ground state $|\Psi_0\rangle$ of a quantum many-body system described by Hamiltonian $H$ can be obtained from a trial wavefunction $|\Psi_T\rangle$ via projection in imaginary time:

$$|\Psi_0\rangle \propto \lim_{\beta \to \infty} \rho(\beta/2) |\Psi_T\rangle \, , \tag{6}$$

where $\rho(\beta) = e^{-\beta H}$ is the imaginary time propagator and $\beta$ is the projection length in imaginary time. Convergence is guaranteed provided $\langle \Psi_0 | \Psi_T \rangle \neq 0$. We expand the trial wavefunction in a complete orthonormal basis $\{|\alpha\rangle\}$ via complex expansion coefficients $C_\alpha \equiv \langle \alpha | \Psi_T \rangle \in \mathbb{C}$:

$$|\Psi_T\rangle = \sum_\alpha C_\alpha |\alpha\rangle \, , \tag{7}$$

chosen to partially diagonalize the Hamiltonian:

$$H = H_0 + H_1 \, , \tag{8}$$

where

$$H_0 |\alpha\rangle = \varepsilon_\alpha |\alpha\rangle \, . \tag{9}$$

In the interaction picture, $\rho(\beta)$ can then be expressed as:

$$\rho(\beta) = e^{-\beta H_0} T_\tau e^{-\int_0^\beta d\tau H_1(\tau)} \, , \tag{10}$$

where $T_\tau$ is the imaginary time-ordering operator and:

$$H_1(\tau) = e^{\tau H_0} H_1 e^{-\tau H_0} \, . \tag{11}$$

The propagator thus admits the expansion:

$$\rho(\beta) = e^{-\beta H_0} \left[ \mathbb{1} - \int_0^\beta d\tau H_1(\tau) + \sum_{Q=2}^\infty (-1)^Q \prod_{q=1}^Q \int_0^{\tau_{q+1}} d\tau_q H_1(\tau_q) \right] \, , \tag{12}$$

where $\mathbb{1}$ is the identity operator, $Q$ the expansion order, and imaginary times are ordered such that $\tau_0 \equiv 0 < \tau_1 < \tau_2 < \cdots < \tau_Q < \tau_{Q+1} \equiv \beta$. Using Eq. (12), the matrix elements of the propagator can be written as

$$\rho(\alpha, \alpha'; \beta) = \left\langle \alpha' \middle| e^{-\beta H} \middle| \alpha \right\rangle = \sum_{Q=0}^{\infty} \rho^{(Q)}(\alpha, \alpha'; \beta), \tag{13}$$

where superscripts denote the order of the term in the expansion. The zeroth-order term reduces to:

$$\rho^{(0)}(\alpha, \alpha'; \beta) = e^{-\epsilon_{\alpha'}\beta} \left\langle \alpha' \middle| \mathbb{1} \middle| \alpha \right\rangle = e^{-\epsilon_\alpha \beta}. \tag{14}$$

Similarly, the first-order term of Eq. (13) becomes:

$$\rho^{(1)}(\alpha, \alpha'; \beta) = -\int_0^\beta d\tau \, e^{-\epsilon_{\alpha'}\beta} \left\langle \alpha' \middle| H_1(\tau) \middle| \alpha \right\rangle = -\int_0^\beta d\tau \, e^{-\epsilon_{\alpha'}(\beta-\tau)} e^{-\epsilon_\alpha \tau} H_1^{\alpha',\alpha}, \tag{15}$$

where we have introduced the notation $H_1^{\alpha',\alpha} \equiv \left\langle \alpha' \middle| H_1 \middle| \alpha \right\rangle$ for the off-diagonal matrix element. Finally, the second and higher-order terms can be simplified by inserting appropriate resolutions of the identity:

$$\begin{aligned}
\rho^{(2)}(\alpha, \alpha'; \beta) &= \int_0^\beta d\tau_2 \int_0^{\tau_2} d\tau_1 \, e^{-\epsilon_{\alpha'}\beta} \left\langle \alpha' \middle| H_1(\tau_2) H_1(\tau_1) \middle| \alpha \right\rangle \\
&= \sum_{\alpha_1} \int_0^\beta d\tau_2 \int_0^{\tau_2} d\tau_1 \, e^{-\epsilon_{\alpha'}(\beta-\tau_2)} e^{-\epsilon_{\alpha_1}(\tau_2-\tau_1)} e^{-\epsilon_\alpha \tau_1} H_1^{\alpha',\alpha_1} H_1^{\alpha_1,\alpha}. 
\end{aligned} \tag{16}$$

Examining the form of Eqs. (14), (15), and (16), we can write the general expansion:

$$\rho(\alpha_0, \alpha_\beta; \beta) = \sum_{Q=0}^{\infty} \sum_{\alpha_1 \ldots \alpha_{Q-1}} \int d\tau_Q \, (-1)^Q e^{-\epsilon_{\alpha_0}\tau_1} \prod_{q=1}^{Q} e^{-\epsilon_{\alpha_q}(\tau_{q+1}-\tau_q)} H_1^{\alpha_q,\alpha_{q-1}}, \tag{17}$$

with the assignment $\alpha_Q \equiv \alpha_\beta$ and $\tau_{Q+1} \equiv \beta$, and we have introduced the short-hand notation:

$$\int d\tau_Q \equiv \int_0^{\tau_{Q+1}} d\tau_Q \int_0^{\tau_Q} d\tau_{Q-1} \cdots \int_0^{\tau_2} d\tau_1. \tag{18}$$

In order to gain physical intuition about the form of the propagator in Eq. (17), we will consider a specific model of bosons on a lattice that will allow us to compute the explicit form of the matrix elements $\epsilon_\alpha$ and $H_1^{\alpha',\alpha}$.

### 3.2 Bose-Hubbard Model

In this paper, the algorithmic developments underlying `PIGSFLI` will be benchmarked on the Bose-Hubbard model describing the dynamics of itinerant bosons on a lattice [85]:

$$H = -t \sum_{\langle i,j \rangle} b_i^\dagger b_j + \frac{U}{2} \sum_i n_i(n_i - 1) - \mu \sum_i n_i. \tag{19}$$

Here, $t$ is the tunneling between neighboring lattice sites $\langle i,j \rangle$, $U > 0$ is a repulsive interaction potential, $\mu$ is the chemical potential, and $b_i^\dagger (b_i)$ are bosonic creation(annhilation) operators on site $i$, satisfying the commutation relation: $\left[ b_i, b_j^\dagger \right] = \delta_{i,j}$, with $n_i = b_i^\dagger b_i$ the local number operator. Since our simulations will be performed in the canonical ensemble, $\mu$ is a simulation

parameter, that we set using the method described in Ref. [86] to improve efficiency by controlling the average number of particles to be around the target value $N$. Appendix A shows $\mu$ being updated in an example simulation.

For this model, it is natural to choose $\{|\alpha\rangle\}$ to be the Fock basis of bosonic number occupation states where

$$|\alpha\rangle = \left| n_0^\alpha, n_1^\alpha \dots, n_{M-1}^\alpha \right\rangle, \tag{20}$$

with $n_i^\alpha$ the number of bosons on site $i$ for the configuration $\alpha$ and $M = L^D$ is the total number of $D$-dimensional hypercubic lattice sites. Then, the kinetic term of the Hamiltonian in Eq. (19) is off-diagonal, whereas the interaction and chemical potential terms constitute the diagonal part:

$$H_0 \equiv \frac{U}{2} \sum_i n_i(n_i - 1) - \mu \sum_i n_i, \tag{21}$$

$$H_1 \equiv -t \sum_{\langle i,j \rangle} b_i^\dagger b_j. \tag{22}$$

Expressing $H_0$ and $H_1$ in the Fock basis yields explicit expressions for the matrix elements:

$$\epsilon_\alpha \equiv \langle \alpha | H_0 | \alpha \rangle = \frac{U}{2} \sum_i n_i^\alpha (n_i^\alpha - 1) - \mu \sum_i n_i^\alpha, \tag{23}$$

$$H_1^{\alpha',\alpha} \equiv \left\langle \alpha' \right| H_1 | \alpha \rangle = -t \sum_{\langle i,j \rangle} \sqrt{n_j^\alpha (n_i^\alpha + 1)} \delta_{n_i^{\alpha'}, n_i^\alpha + 1} \delta_{n_j^{\alpha'}, n_j^\alpha - 1} \prod_{k \notin \{i,j\}} \delta_{n_k^{\alpha'}, n_k^\alpha}. \tag{24}$$

The structure of Eq. (24) ensures that only those off-diagonal elements where $\left| \alpha' \right\rangle$ and $|\alpha\rangle$ differ by one particle hop between different sites will be non-vanishing.

### 3.3 Configuration Space

With access to a representation of the ground state wavefunction via the projection method described in Section 3.1 we can now consider the measurement of ground state expectation values of a physical observable $\mathcal{O}$ in the path integral picture:

$$\begin{aligned}
\langle \mathcal{O} \rangle_0 &\equiv \frac{\langle \Psi_0 | \mathcal{O} | \Psi_0 \rangle}{\langle \Psi_0 | \Psi_0 \rangle} \\
&= \frac{1}{\mathcal{Z}_0} \lim_{\beta \to \infty} \sum_{\alpha_\beta, \alpha'', \alpha', \alpha_0} C_{\alpha_\beta}^* C_{\alpha_0} \rho(\alpha'', \alpha_\beta; \beta/2) \left\langle \alpha'' \middle| \mathcal{O} \middle| \alpha' \right\rangle \rho(\alpha', \alpha_0; \beta/2),
\end{aligned} \tag{25}$$

where we have used the expansion of the trial wavefunction in Eq. (7). The normalization $\mathcal{Z}_0$ can be written as:

$$\mathcal{Z}_0 \equiv \langle \Psi_0 | \Psi_0 \rangle = \lim_{\beta \to \infty} \sum_{\alpha_0, \alpha_\beta} C_{\alpha_\beta}^* C_{\alpha_0} \rho(\alpha_0, \alpha_\beta; \beta) = \lim_{\beta \to \infty} \sum_{Q, \boldsymbol{\alpha}_Q} \int d\boldsymbol{\tau}_Q \, W_0(Q, \boldsymbol{\alpha}_Q, \boldsymbol{\tau}_Q), \tag{26}$$

where we have used Eq. (17) to express the configurational weight:

$$W_0(Q, \boldsymbol{\alpha}_Q, \boldsymbol{\tau}_Q) \equiv C_{\alpha_\beta}^* C_{\alpha_0} (-1)^Q e^{-\epsilon_{\alpha_0} \tau_1} \prod_{q=1}^Q e^{-\epsilon_{\alpha_q}(\tau_{q+1} - \tau_q)} H_1^{\alpha_q, \alpha_{q-1}}, \tag{27}$$

with

$$\boldsymbol{\alpha}_Q \equiv \alpha_0, \alpha_1, \dots, \alpha_{Q-1}, \alpha_\beta, \qquad \boldsymbol{\tau}_Q \equiv \tau_1, \tau_2, \dots, \tau_Q. \tag{28}$$

We can interpret Eq. (26) as an infinite sum over all configurations of particle worldlines connecting state $|\alpha_0\rangle$ at $\tau = 0$ to $|\alpha_\beta\rangle$ at $\tau = \beta$, via $\rho(\alpha_0, \alpha_\beta; \beta)$, with each configuration having statistical weight $W_0$. This formulation can be contrasted with the more commonly studied finite temperature picture where paths are periodic in imaginary time. The expansion order $Q$ in Eq. (26) corresponds to the number of insertions of the off-diagonal operator $H_1$ which (as discussed in Section 3.2) changes the local occupation number of a site (a particle *hop*). These hops can be diagramatically represented as *kinks* in otherwise straight particle worldlines. Fig. 1 shows an example of a worldline configuration for four bosonic particles on four lattice sites.

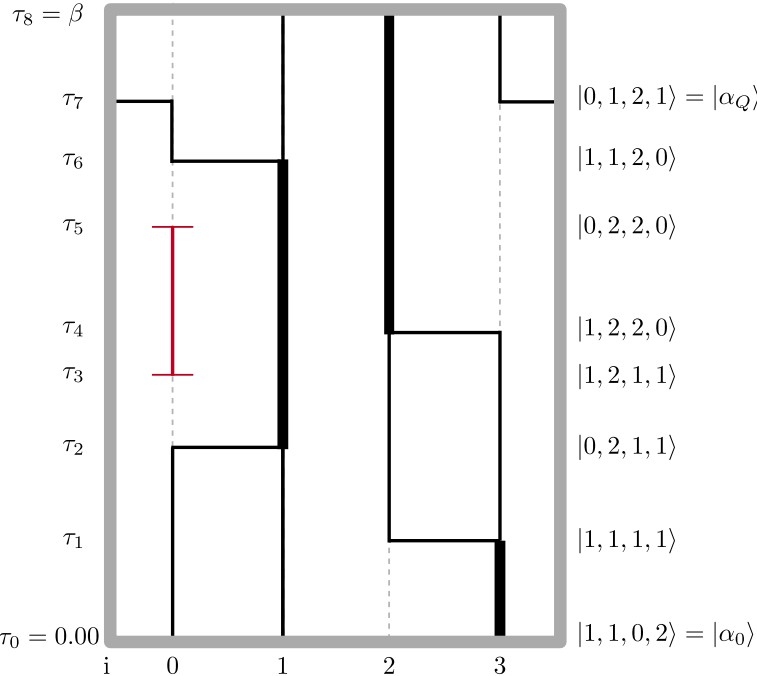

Figure 1: Example of a 4-site worldline configuration. Paths evolve in the direction of imaginary time (vertical axis) and particles can hop between sites (horizontal axis). Imaginary times at which the Fock state changes have been labeled on the left side of the diagram and the corresponding states are shown to the right. Increasing line thicknesses have been used to denote the addition of a particle to that path segment, and a vertical dashed line indicates the absence of a particle. The segment on site $i = 0$ extending from imaginary time $\tau_3$ to $\tau_5$, where a particle was spontaneously created and annihilated later, is called a worm and it will be an integral part of our algorithm. The lattice sites are subject to periodic boundary conditions, as illustrated by the particle hopping from site $i = 3$ to $i = 0$ at imaginary time $\tau_7$.

### 3.3.1 Off-Diagonal Configurations: Worms

A major algorithmic advance in path integral Monte Carlo was attained through the extension of the diagonal configuration space described above to one that includes non-continuous paths (worms), corresponding to insertions of the single particle imaginary time Green function [15]. This technology has also been adapted to continuous space methods [18, 21] and allows for improved sampling performance, extending simulations to $N \simeq 10^4$ particles, as well as allowing for native operation in the grand canonical ensemble. A worm is shown in red in Fig. 1.

Worms can be included in the configuration space through the addition of a source term in the system Hamiltonian:

$$H = H_0 + H_1 \rightarrow H_0 + H_1 + H_{\text{worm}}, \tag{29}$$

where

$$H_{\text{worm}} = -\eta \sum_i (b_i^\dagger + b_i), \tag{30}$$

with $\eta$ the worm fugacity – a tunable parameter associated with the energetic cost of inserting or removing a worm end that can only affect the sampling efficiency. Appendix A shows $\eta$ being updated in an example simulation. Expectation values of physical observables are only accumulated from configurations with no worms presents, however the configurational weight in Eq. (27) needs to be modified:

$$W_0(Q, \boldsymbol{\alpha}_Q, \boldsymbol{\tau}_Q) = C_{\alpha_\beta}^* C_{\alpha_0}(-1)^Q e^{-\epsilon_{\alpha_0} \tau_1} \prod_{q=1}^{Q} e^{-\epsilon_{\alpha_q}(\tau_{q+1} - \tau_q)} \left( H_1^{\alpha_q, \alpha_{q-1}} + H_{\text{worm}}^{\alpha_q, \alpha_{q-1}} \right), \tag{31}$$

where the matrix elements of the source term $H_{\text{worm}}^{\alpha', \alpha}$ can be calculated explicitly in the Fock basis:

$$H_{\text{worm}}^{\alpha', \alpha} = -\eta \sum_i \left( \sqrt{n_i^\alpha + 1} \, \delta_{n_i^{\alpha'}, n_i^\alpha + 1} + \sqrt{n_i^\alpha} \, \delta_{n_i^{\alpha'}, n_i^\alpha - 1} \right) \prod_{k \neq i} \delta_{n_k^{\alpha'}, n_k^\alpha}. \tag{32}$$

## 3.4 Sampling

The Monte Carlo approach to estimating expectation values of observable as in Eq. (25) proceeds by creating a Markov chain from worldline configurations drawn according to the probability density function $\pi(x) = W(x)/\mathcal{Z}$, where $x \equiv \{Q, \boldsymbol{\alpha}_Q, \boldsymbol{\tau}_Q\}$ such that the resulting infinite dimensional sum/integral can be recast as an importance sampling problem. The stochastic rules for transitions having probabilities $T(x \rightarrow x')$ between configurations are independent of the history of the trajectory in state space, and $\pi$ represents the steady state distribution. This is achieved through a set of (possibly pairs of) Monte Carlo updates satisfying detailed balance where $\pi(x)T(x \rightarrow x') = \pi(x')T(x' \rightarrow x)$, as well as fulfilling an ergodicity condition that all configurations are connected by a finite number of steps and there is no periodicity. We factorize the transition probability $T(x \rightarrow x') = P(x \rightarrow x')A(x \rightarrow x')$ into the product of an a priori sampling distribution $P(x \rightarrow x')$ and an acceptance probability $A(x \rightarrow x')$. Combined with detailed balance, the Metropolis-Hastings condition then leads to an expression for the acceptance ratio of a general Monte Carlo update:

$$\frac{A(x \rightarrow x')}{A(x' \rightarrow x)} = \frac{W(x')P(x' \rightarrow x)}{W(x)P(x \rightarrow x')} \equiv R. \tag{33}$$

In [15], the original set of Worm Algorithm updates for finite temperature were introduced. For readers not familiar with them, they will be described in detail in Appendix B. In the remainder of this section, we introduce a new set of $T = 0$ updates that are required due to the breaking of imaginary time translational invariance and the algorithm is then benchmarked on the 1D Bose-Hubbard model in Section 3.6.

## 3.5 New updates for $T = 0$

At finite temperature, both the imaginary time and the spatial direction are subject to periodic boundary conditions, resulting in configurations living on the surface of a $D + 1$ dimensional hypertorus. However, at $T = 0$, only the spatial directions may be periodic and configurations

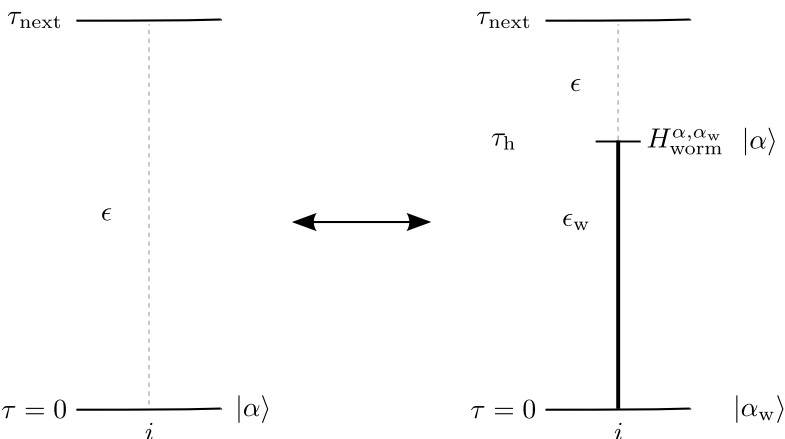

Figure 2: **Insert/Delete worm from** $\tau = 0$. A worm is inserted at the $\tau = 0$ edge by inserting a worm head at a randomly sampled time $\tau_{\mathrm{h}} \in [0, \tau_{\mathrm{next}})$ and adding a particle to the segment spanning the interval $\tau : [0, \tau_{\mathrm{h}})$. An antiworm can instead be inserted by placing a worm tail at the sampled time and destroying a particle in the segment $\tau : [0, \tau_{\mathrm{t}})$.

instead live on a cylinder of length $\beta \to \infty$. As such, the minimum set of updates needed to allow the random walk to visit all worldline configurations is different for $T > 0$ and $T = 0$ simulations. By proposing two new complementary pairs of updates, ergodicity can be satisfied for paths defined on the $\beta$-cylinder.

### 3.5.1  Insert/Delete worm from $\tau = 0$

The first pair of updates required for a $T = 0$ simulation is the insertion and deletion of a worm or antiworm that extends from $\tau = 0$ to some time in the first flat region of the site (i.e., the flat region of the site that has lower bound $\tau = 0$). To insert a worm from $\tau = 0$, a time is randomly sampled within the bounds of the first flat region, $[0, \tau_{\mathrm{next}})$, a worm head is then inserted at the sampled time, and a particle is added to the path segment that extends from $\tau = 0$ to the worm head. For the case of an antiworm, a worm tail is instead inserted at the randomly sampled time, then a particle destroyed from the segment preceding it. If one such worm or antiworm is present in the configuration, its deletion could also be proposed. The update is illustrated in Fig. 2 and proceeds as follows:

**Insert worm from $\tau = 0$:**

0. Attempt update with probability $p_{\mathrm{insertFromZero}}$. This is the bare probability of attempting this move from amongst the entire pool of possible updates. Every other update will have a similar bare attempt probability.

1. Randomly sample site of insertion with probability $1/M$, where $M = L^D$ is the number of total sites.

2. Randomly choose to insert a worm or antiworm with probability $p_{\mathrm{type}}$. If no worm ends are present, sample either worm end with probability $p_{\mathrm{type}} = 1/2$. If there is one end present, select to insert the other type with probability $p_{\mathrm{type}} = 1$.

3. Randomly sample insertion time $\tau_{\mathrm{new}}$ inside the flat region with probability $1/\tau_{\mathrm{next}}$. If a worm has been chosen, then the sampled time corresponds to the time of a worm

head, $\tau_{\text{new}} = \tau_{\text{h}}$. For an antiworm, $\tau_{\text{new}} = \tau_{\text{t}}$ and the update is rejected if there are no particles to destroy in the flat region.

4. Calculate diagonal energy difference $\Delta V \equiv \epsilon_{\text{w}} - \epsilon$, where $\epsilon_{\text{w}}$ is the diagonal energy of the segment of path inside the flat region with more particles, and $\epsilon$ the diagonal energy in the segment with less particles.

5. Sample a random and uniformly distributed number $r \in [0, 1)$.

6. Check,
If $r < R_{\text{insertFromZero}}$, insert worm from $\tau = 0$ into worldline configuration.
Else, reject update and leave worldlines unchanged.

**Delete worm from $\tau = 0$:**

0. Attempt update with probability $p_{\text{deleteFromZero}}$.

1. Randomly choose which of the worm ends present to delete with probability $p_{\text{wormEnd}}$. This can be $1/2$ if there are two worm ends present and $1$ if there is only one worm end.

2. Calculate diagonal energy difference $\Delta V \equiv \epsilon_{\text{w}} - \epsilon$.

3. Sample a random and uniformly distributed number $r \in [0, 1)$.

4. Check,
If $r < R_{\text{deleteFromZero}}$, delete worm from $\tau = 0$ from worldline configuration.
Else, reject update and leave worldlines unchanged.

The acceptance ratios for the insertion from $\tau = 0$ is:

$$
R_{\text{insertFromZero}} = \begin{cases} \frac{C_{\alpha_{\text{w}}}}{C_{\alpha}} \eta \sqrt{n_{\text{w}}} e^{-(\epsilon_{\text{w}} - \epsilon)\tau_{\text{new}}} \frac{p_{\text{deleteFromZero}}}{p_{\text{insertFromZero}}} p_{\text{wormEnd}} \tau_{\text{next}} \frac{M}{p_{\text{type}}} & \text{worm}, \\ \frac{C_{\alpha}}{C_{\alpha_{\text{w}}}} \eta \sqrt{n_{\text{w}}} e^{+(\epsilon_{\text{w}} - \epsilon)\tau_{\text{new}}} \frac{p_{\text{deleteFromZero}}}{p_{\text{insertFromZero}}} p_{\text{wormEnd}} \tau_{\text{next}} \frac{M}{p_{\text{type}}} & \text{anti}, \end{cases}
\tag{34}
$$

and for deletion from $\tau = 0$:

$$
R_{\text{deleteFromZero}} = \frac{1}{R_{\text{insertFromZero}}}.
\tag{35}
$$

Note the appearance of the expansion coefficients of the trial wavefunction $C_{\alpha_{\text{w}}}$ and $C_{\alpha}$, with the former corresponding to the Fock State at $\tau = 0$ with an extra particle on site $i$. For all simulations reported in this work, we have used a constant trial wavefunction, such that the ratio of coefficients becomes unity. The effect of changing the trial wavefunction might be an avenue for further exploration to improve convergence in imaginary time.

### 3.5.2 Insert/Delete worm from $\tau = \beta$

In analogy with insertion/deletion at $\tau = 0$ we also need to consider the opposite end of the $\beta$-cylinder at $\tau = \beta$ (i.e, the flat region bounded from above by $\tau = \beta$). A worm from $\tau = \beta$ is added by inserting a worm tail in the flat region $[\tau_{\text{prev}}, \beta)$, where $\tau_{\text{prev}}$ is the time of the last kink on that site, then adding a particle to the path segment between the worm tail and the end of the flat region at $\tau = \beta$. For an antiworm insertion, a worm head is instead inserted, then a particle destroyed from the path segment between the head and $\tau = \beta$. The update is illustrated in Fig. 3 and proceeds as follows:

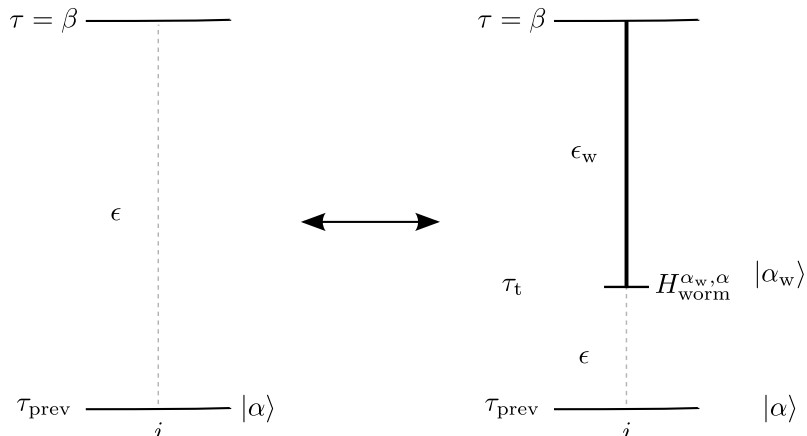

Figure 3: **Insert/Delete worm from** $\tau = \beta$. A worm is inserted at the $\tau = \beta$ edge by inserting a worm tail at a randomly sampled time $\tau_{\mathrm{t}} \in [\tau_{\mathrm{prev}}, \beta)$ and adding a particle to the segment spanning the interval $\tau : [\tau_{\mathrm{t}}, \beta)$. An antiworm can instead be inserted by placing a worm head at the sampled time and destroying a particle in the segment $\tau : [\tau_{\mathrm{h}}, \beta)$.

**Insert worm from $\tau = \beta$:**

0. Attempt update with probability $p_{\mathrm{insertFromBeta}}$.

1. Randomly sample site of insertion with probability $1/M$, where $M$ is the number of total sites.

2. Randomly choose to insert a worm or antiworm with probability $p_{\mathrm{type}}$. If no worm ends are present, sample either worm end with probability $p_{\mathrm{type}} = 1/2$. If there's one end present, select to insert the other type with probability $p_{\mathrm{type}} = 1$.

3. Randomly sample insertion time $\tau_{\mathrm{new}}$ inside the flat region with probability $1/(\beta - \tau_{\mathrm{prev}})$. If a worm has been chosen, then the sampled time corresponds to the time of a worm head, $\tau_{\mathrm{new}} = \tau_{\mathrm{t}}$. For an antiworm, $\tau_{\mathrm{new}} = \tau_{\mathrm{h}}$ and the update is rejected if there are no particles to destroy in the flat region.

4. Calculate diagonal energy difference $\Delta V \equiv \epsilon_{\mathrm{w}} - \epsilon$.

5. Sample a random and uniformly distributed number $r \in [0, 1)$.

6. Check,
If $r < R_{\mathrm{insertFromBeta}}$, insert worm from $\tau = \beta$ into worldline configuration.
Else, reject update and leave worldlines unchanged.

**Delete worm from $\tau = \beta$:**

0. Attempt update with probability $p_{\mathrm{deleteFromBeta}}$.

1. Randomly choose which of the worm ends present to delete with probability $p_{\mathrm{wormEnd}}$. This can be $1/2$ if there are two worm ends present and $1$ if there is only one worm end.

2. Calculate diagonal energy difference $\Delta V \equiv \epsilon_{\mathrm{w}} - \epsilon$.

3. Sample a random and uniformly distributed number $r \in [0, 1)$.

4. Check,
   If $r < R_{\text{deleteFromBeta}}$, delete worm from $\tau = \beta$ from worldline configuration.
   Else, reject update and leave worldlines unchanged.

The acceptance ratios for the insertion/deletion from $\tau = \beta$ update is:

$$
R_{\text{insertFromBeta}} = \begin{cases} \frac{C_{\alpha_{\text{w}}}}{C_\alpha} \eta \sqrt{n_{\text{w}}} e^{-(\epsilon_{\text{w}} - \epsilon)(\beta - \tau_{\text{new}})} \frac{p_{\text{deleteFromBeta}}}{p_{\text{insertFromBeta}}} p_{\text{wormEnd}}(\beta - \tau_{\text{prev}}) \frac{M}{p_{\text{type}}} & \text{worm}, \\ \frac{C_\alpha}{C_{\alpha_{\text{w}}}} \eta \sqrt{n_{\text{w}}} e^{+(\epsilon_{\text{w}} - \epsilon)(\beta - \tau_{\text{new}})} \frac{p_{\text{deleteFromBeta}}}{p_{\text{insertFromBeta}}} p_{\text{wormEnd}}(\beta - \tau_{\text{prev}}) \frac{M}{p_{\text{type}}} & \text{anti}, \end{cases} \tag{36}
$$

$$
R_{\text{deleteFromBeta}} = \frac{1}{R_{\text{insertFromBeta}}} . \tag{37}
$$

The new moves described above, in combination with the original Worm Algorithm moves described for reference in Appendix B, will allow for an ergodic PIMC simulation on the lattice at $T = 0$. In practice we weight all update attempt probabilities equally such that $p_{\text{updateType}} = 1/N_{\text{updates}}$ but these could be optimized to improve simulation efficiency.

## 3.6 Energy benchmarks

To test the validity of the PIGSFLI algorithm, ground state energies have been estimated in a one dimensional Bose-Hubbard model consisting of 8 sites that is amenable to an exact solution. The ground state expectation value of the total energy:

$$
\langle H \rangle_0 \simeq \langle H_0 \rangle_{\text{MC}} + \langle H_1 \rangle_{\text{MC}} , \tag{38}
$$

where the subscript MC indicates a Monte Carlo average over the weighted configuration space of worldlines. The potential energy estimator $\langle H_0 \rangle_{\text{MC}}$ is derived in Appendix C and can be calculated by averaging the on-site interaction over an imaginary time window of size $\Delta\tau$, centered around $\tau = \beta/2$:

$$
\langle H_0 \rangle_{\text{MC}} = \frac{1}{\Delta\tau} \left\langle \int_{\beta/2 - \Delta\tau}^{\beta/2 + \Delta\tau} \frac{U}{2} \sum_i n_i(\tau)(n_i(\tau) - 1) \right\rangle_{\text{MC}} , \tag{39}
$$

where the chemical potential contribution has been neglected as simulations have been performed in the canonical ensemble. The subscript MC has been added to distinguish Monte Carlo averages from usual quantum mechanical expectation values. The kinetic energy estimator is derived in Appendix D and is given by

$$
\langle H_1 \rangle_{\text{MC}} = -\frac{\langle N_{\text{kinks}} \rangle_{\text{MC}}}{\Delta\tau} , \tag{40}
$$

where $\langle N_{\text{kinks}} \rangle_{\text{MC}}$ is average number of kinks inside an imaginary time window of width $\Delta\tau$, centered around $\tau = \beta/2$.

Both of these estimators become stochastically exact in the limit $\beta \to \infty$ and a suitable extrapolation procedure is described in Appendix E. In the limit $\beta \to \infty$, the energy estimators are independent of the window size $\Delta\tau$. For finite $\beta$ there will be additional systematic error from using a larger $\Delta\tau$ due to lack of convergence to the ground state; decreasing $\Delta\tau$ will generally increase the statistical errors due to a reduction in the imaginary time averaging. To balance these competing effects we set the window size to $0.2\beta$.

Fig. 4 shows the relative error between the exact and estimated ground state kinetic and potential energies, respectively, for a one-dimensional Bose-Hubbard lattice of $L = 8$ sites at unit-filling as a function of $\beta t$ for three different values of the on-site repulsion $U/t = 0.5, 3.3, 10$

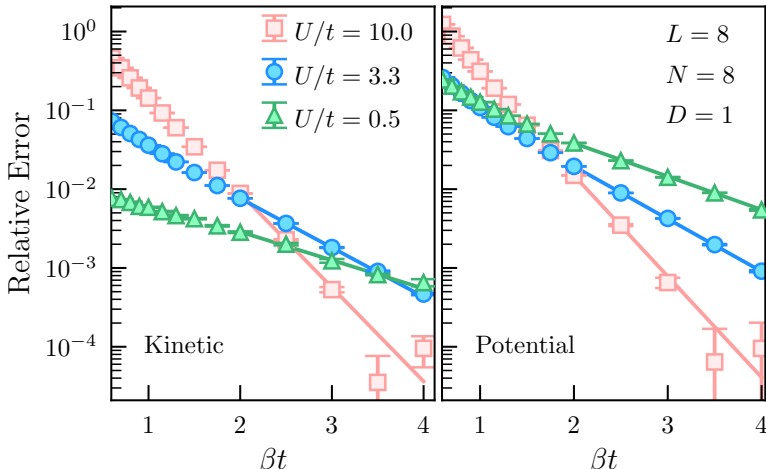

Figure 4: Relative error of the ground state kinetic energy (**left**) and potential energy (**right**) as a function of $\beta t$ for a Bose-Hubbard chain of $L = 8$ sites at unit-filling. As $\beta$ increases, the relative error decays exponentially to zero, as evidenced by the fits (solid lines). Regimes where the interaction strength is large possess a sizeable energy gap and more accurate results can be obtained at lower $\beta t$ values, since the ground state is projected out of the trial wavefunction much faster. The different shapes and colors correspond to different interaction strengths. The interactions strengths $U/t = 0.5, 3.3, 10.0$ represent values in the superfluid phase, the 1D critical point, and the Mott phase, respectively. The measurement window is $\Delta\tau = 0.2\beta t$.

corresponding to the superfluid, critical point, and insulating phases. We would like to mention that the value of the quantum critical point for this system has been extensively studied throughout the years, with various methods giving slightly different estimates for it [87–102]. It is customary to report the interaction strengths in the dimensionless form $U/t$, which is the reason why the projection length is rescaled as $\beta \to \beta t$. In practice, we set the tunneling parameter to $t = 1.0$ for all simulations and only adjust the potential $U$ and projection length $\beta$. In all of these regimes, the relative error decays exponentially as a function of $\beta$ as indicated by solid lines corresponding to the form:

$$E^{\text{err}}(\beta) = C_1 e^{-\beta t C_2} \equiv C_1 e^{-\beta C_2}\,, \tag{41}$$

where $C_1$ and $C_2$ are fitting parameters, $E^{\text{err}}$ denotes the relative error in either of the energies. Notice that for $U/t = 10.0$, the relative error in the energies becomes so small for the largest $\beta$ values that it becomes difficult to resolve the error bars on this scale to high accuracy. This is due to the fact that in the presence of a finite energy gap, the exact ground state can be projected out from the trial wavefunction much faster via Eq. (6). Conversely, near the superfluid phase, where interactions are low, the energy gap is much smaller and the error decays slowly in comparison. Formally, the superfluid phase will be gapless in the thermodynamic limit. In this regime the gap scales polynomially in the system size and thus we must increase $\beta$ accordingly to identify the exponential behavior of observables as a function of projection length. This behavior is also expected when computing other ground state expectation values.

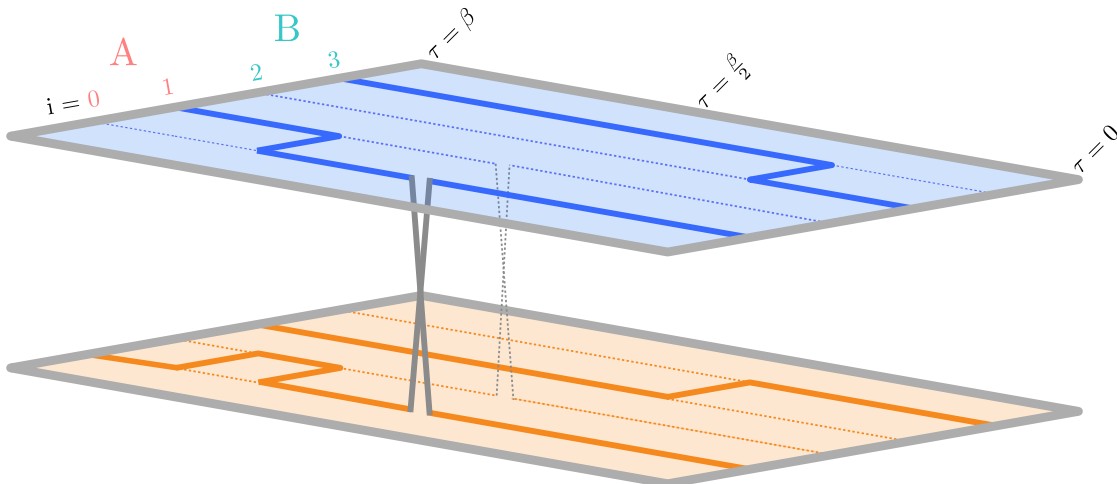

Figure 5: Worldlines in the two-replica space $\alpha = 2$ in which Rényi entanglement entropies can be measured. A special type of SWAP kink between replicas can be inserted or deleted at $\tau = \beta/2$ on the sites that belong to subregion $A$. In this example, two SWAP kinks are shown on sites $i = 0$ and $i = 1$.

# 4 Rényi entanglement entropies from SWAP

With a working Path Integral Monte Carlo algorithm at $T = 0$ now at hand, the next goal will be to introduce a method to perform estimates of quantum entanglement in the Bose-Hubbard model under a spatial bipartition. This approach will allow for the investigation of the entanglement properties of much larger systems than those that can be studied with exact diagonalization and is based on extensive previous algorithmic development in quantum Monte Carlo based on the replica trick [3, 5–8, 10, 11, 103–113]. The goal is to recast the measurement of the Rényi entanglement entropy in terms of a local expectation value of an operator that can be sampled with our Monte Carlo method.

## 4.1 Entanglement entropy estimator

We can compute the Rényi entanglement entropy in quantum Monte Carlo by performing simulations of two (or more) identical and non-interacting copies of the system. For $\alpha = 2$, we'll consider the system $|\Psi\rangle$ and a replica $\left|\tilde{\Psi}\right\rangle$, and note that $S_2(\rho_A)$ can be related to the expectation value of the unitary $\mathsf{SWAP}_A$ operator [3] that acts on the replicated Hilbert space. An example replicated worldline configuration is shown in Fig. 5.

Defining $\{|a\rangle\}$ and $\{|b\rangle\}$ to be bases of states that are localized to subregion A and its complement, respectively, the $\mathsf{SWAP}_A$ operator is defined on the tensor product states $|a, b\rangle \equiv |a\rangle \otimes |b\rangle$ to exchange the states of the subsystem A between the replicas.:

$$\mathsf{SWAP}_A\left[|a, b\rangle \otimes \left|\tilde{a}, \tilde{b}\right\rangle\right] \equiv \left|\tilde{a}, b\right\rangle \otimes \left|a, \tilde{b}\right\rangle . \tag{42}$$

For an arbitrary state $|\Psi\rangle = \sum_{a,b} C_{ab} |a,b\rangle$, the expectation value of $\mathrm{SWAP}_A$ takes the following form:

$$
\begin{aligned}
\langle \mathrm{SWAP}_A \rangle &\equiv \langle \Psi_0 | \otimes \langle \tilde{\Psi}_0 | \, \mathrm{SWAP}_A \, |\Psi_0\rangle \otimes |\tilde{\Psi}_0\rangle \\
&= \langle \Psi_0 | \otimes \langle \tilde{\Psi}_0 | \sum_{a,b} C_{ab} \sum_{\tilde{a},\tilde{b}} C_{\tilde{a}\tilde{b}} |\tilde{a},b\rangle \otimes |a,\tilde{b}\rangle = \sum_{a,\tilde{a}} \left( \sum_b C_{\tilde{a}b}^* C_{ab} \right) \left( \sum_{\tilde{b}} C_{a\tilde{b}}^* C_{\tilde{a}\tilde{b}} \right) \\
&= \sum_{a,\tilde{a}} \rho_A(\tilde{a},a) \rho_A(a,\tilde{a}) = \mathrm{Tr}\, \rho_A^2 ,
\end{aligned}
$$

where $\rho_A(\tilde{a},a) = \langle \tilde{a} | \rho_A | a \rangle$ are elements of the reduced density matrix.

The second Rényi EE can then be computed from the expectation value of $\mathrm{SWAP}_A$ as

$$
S_2(\rho_A) = -\ln\langle \mathrm{SWAP}_A \rangle . \tag{43}
$$

To measure the $\mathrm{SWAP}_A$ estimator, two non-interacting replicas of the worldline configuration space are sampled. The sampling weight of these statistically independent sets of worldlines is the product of their weights $W(Q, \boldsymbol{\alpha}_Q, \boldsymbol{\tau}_Q)\tilde{W}(\tilde{Q}, \tilde{\boldsymbol{\alpha}}_Q, \tilde{\boldsymbol{\tau}}_Q)$, where the tilde vs non-tilde refers to quantities in different replicas. For the measurement of entanglement, the sampled ensemble also allows for the possibility of kinks occurring at $\tau = \beta/2$ that connect the spatial subregion $A$ of each of the replicas.

In the replicated configuration space, the ground state can be projected out of a trial wavefunction by generalizing the projection relation in Eq. (6) as:

$$
|\Psi_0\rangle \otimes |\tilde{\Psi}_0\rangle = \lim_{\beta \to \infty} e^{-\frac{\beta}{2} H \otimes \mathbb{1}} |\Psi_T\rangle \otimes e^{-\frac{\beta}{2} \mathbb{1} \otimes H} |\tilde{\Psi}_T\rangle . \tag{44}
$$

Where the operator structure reflects operation on the system (replica).

$$
\begin{aligned}
\langle \mathrm{SWAP}_A \rangle = \lim_{\beta \to \infty} \sum_{\substack{\alpha_0, \alpha_{\beta/2-}\\ \tilde{\alpha}_0, \tilde{\alpha}_{\beta/2-}}} \sum_{\substack{\alpha_{\beta/2+}, \alpha_\beta\\ \tilde{\alpha}_{\beta/2+}, \tilde{\alpha}_\beta}} C_{\alpha_\beta}^* C_{\tilde{\alpha}_\beta}^* C_{\alpha_0} C_{\tilde{\alpha}_0} \rho(\alpha_\beta, \alpha_{\beta/2+}; \beta/2) \tilde{\rho}(\tilde{\alpha}_\beta, \tilde{\alpha}_{\beta/2+}; \beta/2) \\
\times \langle \alpha_{\beta/2+} \otimes \tilde{\alpha}_{\beta/2+} | \, \mathrm{SWAP}_A \, |\alpha_{\beta/2-} \otimes \tilde{\alpha}_{\beta/2-}\rangle \rho(\alpha_{\beta/2-}, \alpha_0; \beta/2) \tilde{\rho}(\tilde{\alpha}_{\beta/2-}, \tilde{\alpha}_0; \beta/2), \quad (45)
\end{aligned}
$$

where $\alpha_{\beta/2-}$ and $\alpha_{\beta/2+}$ denote the Fock state immediately before and after $\beta/2$, respectively. Defining the bipartitioned Fock states $|\alpha\rangle = |a,b\rangle$

$$
\begin{aligned}
\langle \alpha_{\beta/2+} \otimes \tilde{\alpha}_{\beta/2+} | \, \mathrm{SWAP}_A \, |\alpha_{\beta/2-} \otimes \tilde{\alpha}_{\beta/2-}\rangle \\
= \langle a_{\beta/2+}, b_{\beta/2+} \otimes \tilde{a}_{\beta/2+}, \tilde{b}_{\beta/2+} | \tilde{a}_{\beta/2-}, b_{\beta/2-} \otimes a_{\beta/2-}, \tilde{b}_{\beta/2-}\rangle \\
= \delta_{a_{\beta/2+}, \tilde{a}_{\beta/2-}} \delta_{\tilde{a}_{\beta/2+}, a_{\beta/2-}} .
\end{aligned} \tag{46}
$$

Where the Kronecker-Delta functions are understood as the product of individual $\delta$-functions over the sites in spatial subregion $A$. The ground state expectation value is then given by:

$$
\begin{aligned}
\langle \mathrm{SWAP}_A \rangle = \lim_{\beta \to \infty} \sum_{Q_-, \boldsymbol{\alpha}_{Q_-}} \int d\boldsymbol{\tau}_{Q_-} \, W_0(Q_-, \boldsymbol{\alpha}_{Q_-}, \boldsymbol{\tau}_{Q_-}) \sum_{Q_+, \boldsymbol{\alpha}_{Q_+}} \int d\boldsymbol{\tau}_{Q_+} \, W_0(Q_+, \boldsymbol{\alpha}_{Q_+}, \boldsymbol{\tau}_{Q_+}) \\
\times \sum_{\tilde{Q}_-, \tilde{\boldsymbol{\alpha}}_{\tilde{Q}_-}} \int d\tilde{\boldsymbol{\tau}}_{\tilde{Q}_-} \, \tilde{W}_0(\tilde{Q}_-, \tilde{\boldsymbol{\alpha}}_{\tilde{Q}_-}, \tilde{\boldsymbol{\tau}}_{\tilde{Q}_-}) \sum_{\tilde{Q}_+, \tilde{\boldsymbol{\alpha}}_{\tilde{Q}_+}} \int d\tilde{\boldsymbol{\tau}}_{\tilde{Q}_+} \, \tilde{W}_0(\tilde{Q}_+, \tilde{\boldsymbol{\alpha}}_{\tilde{Q}_+}, \tilde{\boldsymbol{\tau}}_{\tilde{Q}_+}) \\
\times \left( \delta_{a_{\beta/2+}, \tilde{a}_{\beta/2-}} \delta_{\tilde{a}_{\beta/2+}, a_{\beta/2-}} \right) .
\end{aligned} \tag{47}
$$

The expression above is in the form of a statistical average over paths of the product of $\delta$-functions $\delta_{a_{\beta/2+},\tilde{a}_{\beta/2-}}\delta_{\tilde{a}_{\beta/2+},a_{\beta/2-}}$, up to a normalization factor. The estimator of the expectation value of the $\text{SWAP}_A$ operator finally becomes:

$$\langle \text{SWAP}_A \rangle_{\text{MC}} = \langle \delta_{a_{\beta/2+},\tilde{a}_{\beta/2-}}\delta_{\tilde{a}_{\beta/2+},a_{\beta/2-}} \rangle_{\text{MC}} \,. \tag{48}$$

In practice, this expectation value can be computed by building a histogram of the number of times each possible number of SWAP kinks was measured. This number of SWAP kinks will range from 0 to some maximum number $m_A$. One then takes the bin corresponding to the desired spatial partition size, and normalizes it by dividing by the bin corresponding to zero SWAP kinks measured. Histograms are only updated when both replicas have the same number of total particles.

## 4.2 Symmetry-resolved entanglement

Consider the projection of a single-replica ground state onto the subspace of fixed local particle number $n$:

$$\left| \Psi_0^{(n)} \right\rangle = \Pi_n \left| \Psi_0 \right\rangle = \sum_{a^{(n)}} C_{a^{(n)},b^{(N-n)}} \left| a^{(n)}, b^{(N-n)} \right\rangle , \tag{49}$$

where $\Pi_n$ is the projection operator defined in Eq. (4), $a^{(n)}$ are configurations in $A$ with $n$ particles, and $b^{(N-n)}$, configurations in $B$ containing the remaining $N-n$ particles. It is shown in Appendix F that this projected ground state can be replicated and then taking the expectation value of the $\text{SWAP}_A$ operator gives:

$$\langle \text{SWAP}_{A_n} \rangle \equiv \left\langle \Psi_0^{(n)} \otimes \tilde{\Psi}_0^{(n)} \right| \text{SWAP}_A \left| \Psi_0^{(n)} \otimes \tilde{\Psi}_0^{(n)} \right\rangle = \mathcal{N}^{-1} \text{Tr}\, \rho_{A_n}^2 , \tag{50}$$

where $\mathcal{N}$ is a normalization constant. The symmetry-resolved entanglement entropy for the ground state projected onto a fixed local particle number sector can be computed from a Monte Carlo estimator for $\langle \text{SWAP}_{A_n} \rangle$; this can be obtained similarly to the estimator Eq. (48). That is, by building a histogram of the number of times that each possible number of $\text{SWAP}_A$ kinks is measured. One such histogram needs to be built for every possible local particle number $n$ in the $A$ subregion. The reciprocal of the normalization constant is just the weight of the replicated ground state consisting of both replicas having local particle number $n$ in subregion $A$. This can be estimated by accumulating the number of times that replicated configurations with zero SWAP kinks and the same local particle number $n$ in both replicas were measured.

## 4.3 Operationally accessible entanglement

The accessible entanglement entropy, Eq. (5), and for $\alpha = 2$, can be written in terms of expectation values of $\text{SWAP}_A$ operators

$$S_2^{\text{acc}}(\rho_A) = -2\ln\left[ \sum_n P_n [\text{Tr}\, \rho_{A_n}^2]^{1/2} \right] = -2\ln\left[ \sum_n P_n \left[ \mathcal{N}\langle \text{SWAP}_{A_n} \rangle \right]^{1/2} \right] . \tag{51}$$

In addition to $\langle \text{SWAP}_{A_n} \rangle$, which can be computed as described in the previous section, the local particle number probability distribution $P_n$ is estimated by building a histogram of $n$, the number of particles in subsystem $A$.

The accessible entanglement may also be computed directly from the full Rényi entanglement entropy and the subsystem particle number distribution [33]:

$$S_\alpha^{\text{acc}} = S_\alpha - H_{1/\alpha}\left( \{ P_{n,\alpha} \} \right) , \tag{52}$$

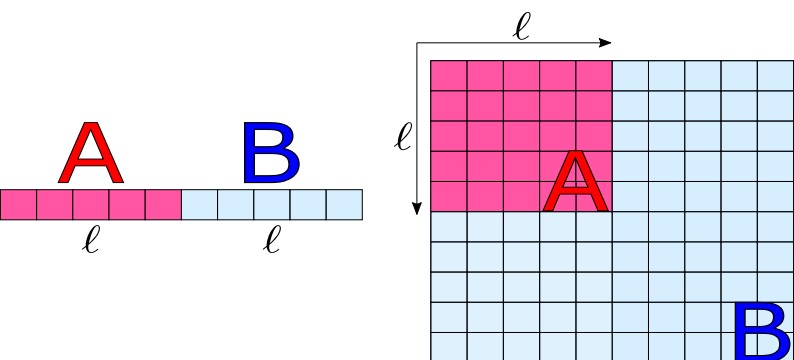

Figure 6: Example systems: **(left)** 1D chain under equal spatial bipartitions of size $\ell$ and **(right)** a square lattice with square spatial subregion of linear size $\ell$. The total number of sites in the subregions is $m \equiv \ell^D$. Periodic boundary conditions are used in all cases.

where $H_\alpha(\{P_n\})$ is the Rényi generalization of the Shannon entropy of $P_n$:

$$H_\alpha\big(\{P_n\}\big) = -\frac{1}{\alpha - 1} \ln \sum_n P_n^\alpha. \tag{53}$$

Thus the accessible entanglement entropy can be computed either from measurement of the symmetry resolved entanglement entropies or measurements of the full entanglement entropy, once $P_n$ has been computed. Note that Eq. (51) might yield undefined results because $P_n$ or $\langle \mathrm{SWAP}_{A_n} \rangle$ could be zero for local particle number sectors with small probability. Appendix G describes a numerical scheme in which contributions from these low probability sectors are discarded so a well-behaved and accurate estimate for Eq. (51) can be obtained.

Now that the estimators of the Rényi entanglement entropy have been defined, we will describe below additional configurational Monte Carlo updates required for their measurement.

## 5 Entanglement entropies via `PIGSFLI`

In order to measure estimators for the Rényi entanglement entropy (Eq. (43)), the symmetry-resolved entanglement entropy , and the operationally accessible entanglement entropy (Eq. (51)), the simulation configuration space needs to be modified to include replicated word-lines [3] and additional updates are needed to sample the insertion of connections (SWAP kinks) between them.

### 5.1 SWAP Updates

#### 5.1.1 Insert/Delete SWAP kink

The first pair of updates that need to be added is to insert/delete a pair of kinks, one for each replica, at $\tau = \beta/2$ that connects worldlines between replicas. This pair of kinks can only originate/terminate from lattice sites inside subregion $A$. SWAP kinks are only inserted or deleted whenever the number of particles in the site at $\tau = \beta/2$ is the same for both replicas.

The update is illustrated in Fig. 7 and proceeds as follows:

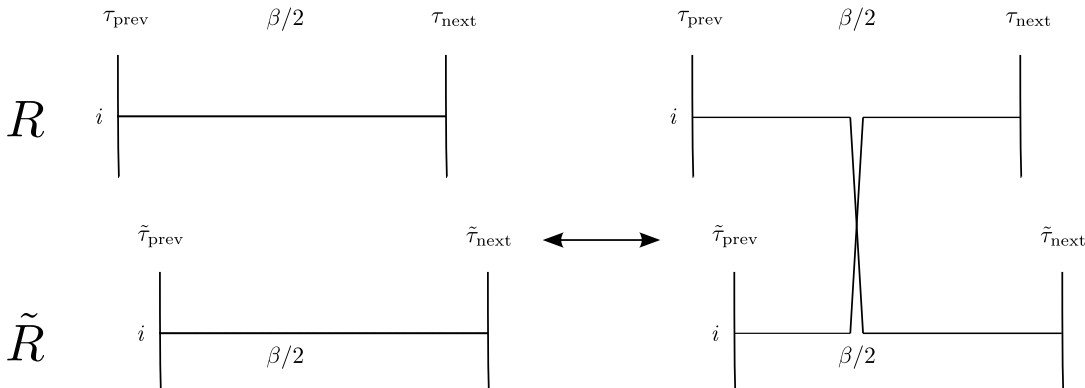

Figure 7: Insert/Delete SWAP kink. The number of particles at the center of the path, $\tau = \beta/2$, is measured for the same site on the two different replicas. If the number of particles is the same, then the SWAP kink is inserted. The kinks are shown to form an "X" in the diagram for visual clarity, but they both exist at exactly $\beta/2$. Kink deletion occurs if the number of particles at $\beta/2$ is the same for site $i$ of both system and replica. In the figure, the replicas are labeled $R$ and $\tilde{R}$.

**Insert SWAP kink:**

0. Attempt update with probability $p_{\mathrm{insertSWAPKink}}$.

1. Systemically choose a subregion site that has no kinks and get particle number on the site at $\tau = \beta/2$ in both replicas. There is no unique way of systemically choosing the site. In 1D one can, for example, always choose to insert at site $i + 1$ if site $i$ already has a SWAP kink. In 2$D$, where the subregion is also a square, one can propose insertions on a row $i$ in the same way as the 1D case, and when full, move to the next row $i + 1$, then $i + 2$, etc ... Fig. 6 shows example lattices in one and two dimensions, with the subregion in which SWAP kinks will be inserted in pink.

2. Insert SWAP kink at $\tau = \beta/2$ with unity acceptance rate if the on-site particle number at $\tau = \beta/2$ is the same for both replicas.

**Delete SWAP kink:**

0. Attempt update with probability $p_{\mathrm{deleteSWAPKink}}$.

1. Choose site at which last SWAP kink was inserted and get particle number on the site at $\tau = \beta/2$ in both replicas.

2. Delete SWAP kink at $\tau = \beta/2$ with unity acceptance rate if the on-site particle number at $\tau = \beta/2$ is the same for both replicas.

The reason that the acceptance rate is unity for these updates is due to the restriction that local particle number be the same on both replicas at the SWAP kink insertion/deletion site. Since the number of particles will be unchanged at any path segment, there is no energetic difference for configurations post and pre update and the ratio of configurational weights post and pre update is one: $W'/W = 1$. The probability ratio of proposing a SWAP deletion to SWAP insertion also is unity: $P(x' \to x)/P(x \to x') = 1$. This is due to the systematic way in which the insertion and deletion sites are chosen. Taking the product of both ratios, then the Metropolis acceptance ratio is also unity: $R = 1$.

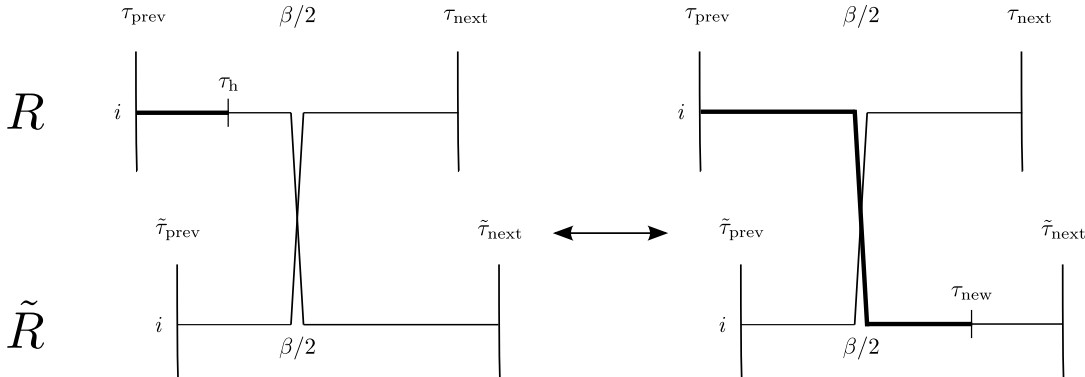

Figure 8: Advance/Recede along SWAP kink. If a worm end, either head or tail is adjacent to a SWAP kink, it can be shifted in the imaginary time direction and moved to the other replica if the new randomly sampled time goes across $\beta/2$. The diagram above shows the example of advancing/receding a worm head, at time $\tau_{\mathrm{h}}$, along a SWAP kink and moved to the other replica to a new time, $\tau_{\mathrm{new}}$.

### 5.1.2 Advance/Recede along SWAP kink

This update can be seen in Fig. 8 and is a direct generalization of the advance/recede move of the original Worm Algorithm updates (see Section B.2) to the case where a worm end is moved across a SWAP kink connecting the system and replica. Thus, the upper and lower imaginary time bounds of the flat interval will now be in different replicas. And in the same way as its single-replica counterpart, the new time of the worm end ($\tau_{\mathrm{new}}$) is sampled from the truncated exponential distribution in Eq. (B.5) to yield an acceptance ratio of unity.

The PIGSFLI algorithm has been now fully described. In the next section, entanglement entropy, symmetry resolved entanglement entropy and operationally accessible entanglement results obtained with the algorithm are presented.

## 6 Results

Previous numerical studies of entanglement in the Bose-Hubbard model have mostly focused on small system sizes using exact diagonalization [34, 36, 114, 115] or matrix product based methods [87, 116–119] which enforce an occupation restriction on the local Hilbert space for soft-core bosons. Results in two spatial dimensions exist [120–122], but they are more scarce, especially for the symmetry resolved and accessible entanglements.

We begin by benchmarking the capability of the PIGSFLI algorithm for entanglement quantification. The relative error of the second Rényi entanglement entropy for a small system of $L = M = 8$ in one dimension as a function of the projection length $\beta$ is shown in Fig. 9 for a maximal spatial bipartition with $\ell = L/2 = 4$. Here, the error is calculated using the exact result via a ground state diagonalization of the full Hamiltonian. The Monte Carlo estimates have been obtained by averaging over many seeds and using the jackknife method for error bar estimation.

We report results for interaction strengths $U/t = 0.5, 3.3, 10.0$, characteristic of the superfluid phase, the critical point, and Mott insulating phases, respectively. Due to the small energy gap in the superfluid phase, it is seen that the exact result is projected out via QMC at a much slower rate when increasing $\beta$ than compared to regimes where the energy gap is large. However, for all three interaction strengths considered, good accuracy is achieved, with $< 1\%$ relative error.

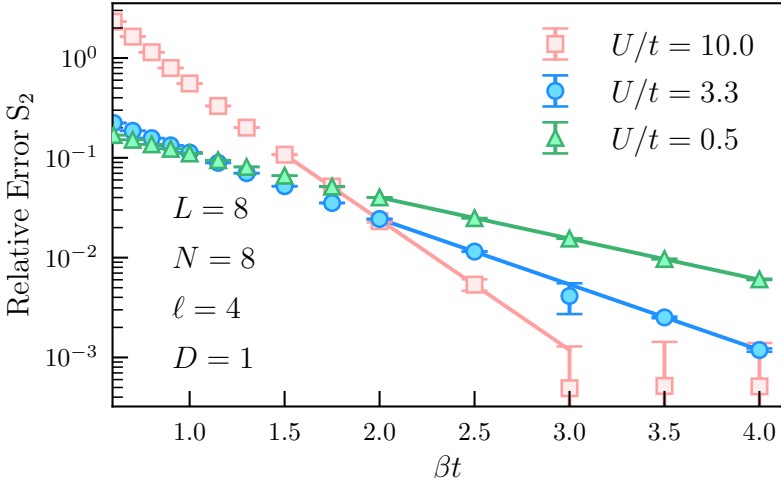

Figure 9: $\beta$-scaling of relative error of full entanglement $S_2$ for a 1D lattice of $L = 8$ sites at unit-filling. The system is bipartitioned into equally sized subregions $A, B$ of size $\ell = 4$ sites. The relative error of $S_2$ decays as a function of $\beta t$. The solid lines are simple exponential fits.

Moving to larger system sizes, and verifying how entanglement changes at quantum phase transitions, we show the second Rényi entropy in Fig. 10 across the phase diagram. The left panel displays results for a 1D Bose-Hubbard chain of $L = 16$ sites at unit-filling, under an equal spatial bipartition composed of $\ell = 8$ sites. For this 16 particle system, exact diagonalization can still be employed, and the exact result is included as a solid line. QMC results are obtained for a range of projection lengths $\beta$ at each interaction strength. At small interactions $U/t \ll 1$, deep in the superfluid phase, and up to a value of $U/t \approx 10$, the systematic error falls as $\beta$ increases. Deeper in the insulating phase, even though in principle this exponential decay of the systematic error is still happening, all data points are seen to collapse onto the exact result on this scale. This is again due to the finite energy gap causing the exact ground state expectation values to be projected out much faster (i.e., for smaller $\beta$) from the trial (constant) wavefunction. Improved projection behavior could be obtained by tuning the trial state as a function of $U/t$.

The $\beta \to \infty$ asymptotic value of $S_2$ can be systematically obtained by performing a three-parameter exponential fit of QMC data to the form:

$$S_2(\beta) = S_2^{(\beta \to \infty)} + C_1 e^{-C_2 \beta} , \tag{54}$$

where $S_2^{(\beta \to \infty)}$, $C_1$, and $C_2$ are fitting parameters. The extrapolated second Rényi entanglement entropies are shown as solid circles. All extrapolations were observed to fall within one standard deviation of the exact result, and the range of $\beta$ needed to access this exponential scaling regime is dependent both on interaction strength and system size. Appendix E includes additional details on the fitting process.

The right panel of Fig. 10 shows the same interaction sweep, but for a 1D Bose-Hubbard ring of $L = 256$ lattice sites at unit-filling under an equal spatial bipartition of $\ell = 128$ sites. Here we only include results extrapolated to $\beta \to \infty$. For this larger system size, the phase transition is more clearly seen near its thermodynamic limit value $(U/t)_c \approx 3.3$ [87–102] via an accompanying reduction in the spatial entanglement as signature of the adjacent insulating phase for strong repulsive interactions. In this regime, due to unit-filling, the ground state approaches a product state with one localized boson per site; thus the entanglement vanishes as $U/t \to \infty$. The behavior of $S_2$ across the transition sweep is similar to what was seen

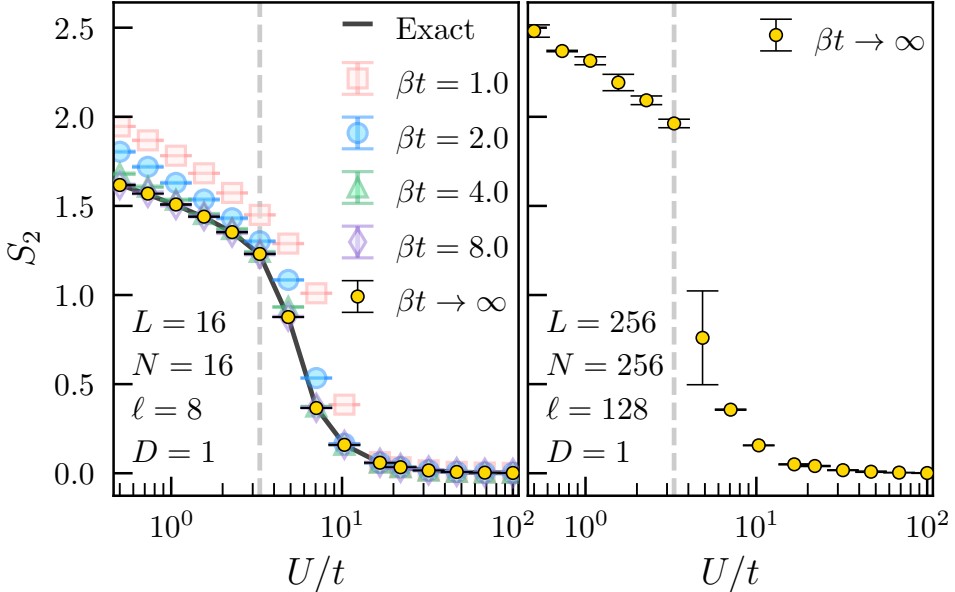

Figure 10: **(left)** Full Rényi Entanglement Entropy $S_2$ for a 1D Bose-Hubbard lattice with $L = 16$ sites at unit-filling under an equal spatial bipartition of size $\ell = 8$ at various interaction strengths $U$. The entropies were measured from simulations at four different values of $\beta t$. As expected, the exact ground state value (solid black line) is approached as $\beta t$ increases. The vertical dashed line at $U = 3.3$ is the exact value of the 1D Superfluid-Mott Insulator phase transition. The solid circles are large $\beta t$ extrapolations of $S_2$ obtained from a three-parameter exponential fit of $S_2$ results at $\beta t = 4, 6, 8, 10, 12$. **(right)** Full Rényi Entanglement Entropy $S_2$ for a 1D Bose-Hubbard lattice with $L = 256$ sites at unit-filling under an equal spatial bipartition of size $\ell = 128$ at various interaction strengths $U/t$.

in Ref. [12] for $^{87}$Rb atoms on a $L = 4$ site lattice. Access to larger systems sizes opens the window for an accurate determination of the quantum critical point via a finite size scaling analysis using the second Rényi entropy, as it was done in Ref. [101] using a measurement based on the von Neumann entropy.

One of the main benefits of our QMC approach is that it can be easily adapted to general spatial dimension $D$, with spatial connections (e.g. hopping or interactions) in the Hamiltonian being encoded through an adjacency matrix. In Fig. 11, we show the scaling of the $\alpha = 2$ Rényi entanglement entropy for the two dimensional Bose-Hubbard model with linear size $L = 32$, corresponding to $M = 32 \times 32 = 1024$ total sites at unit-filling. Using the extrapolation method discussed above (and in Appendix E), $S_2$ was determined as a function of $\ell$, the linear size of a square subregion. QMC calculations were performed at a single value of the interaction, $U/t \approx 16.7$, near the $2D$ critical point [90, 123, 124]. Subregions with linear sizes $\ell = 1, \ldots, 20$ were investigated, and we observe the scaling $S_2 \sim \ell$ as expected due to the presence of an entanglement area law [43, 44, 125, 126]. We fit the entanglement to the general scaling form:

$$S_2(\ell) = a\ell + b \ln \ell + c, \tag{55}$$

where we ignore corrections of $\mathcal{O}(1/\ell)$, which we are unable to resolve with our current dataset. By subtracting off the dominant linear term in $\ell$ scaling after fitting, we can investigate the sub-leading logarithmic correction as a function of subsystem linear size $\ell$, with the results shown in the inset of Fig. 11. In systems with a continuous symmetry breaking in the thermodynamic limit, the logarithmic correction, $b$, and constant, $c$, of Eq. (55) can con-

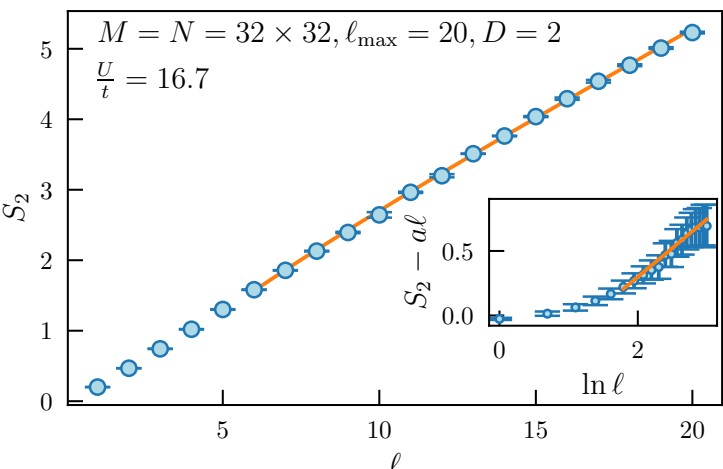

Figure 11: Finite-size scaling of the second Rényi Entropy in a square lattice of size $M = 32 \times 32$ at unit-filling for various subregion sizes. The subregions are made up of lattice sites arranged as squares of linear sizes $\ell = 1, 2, \ldots, 20$. The entanglement is seen to increase linearly with the boundary of the subregion. The data is fit to a linear model with a sub-leading correction term that is logarithmic in $\ell$, as shown in Eq. (55), yielding $a \approx 0.2$, $b \approx 0.5$, and $c \approx -0.6$. The interaction strength was fixed to a value near the $2D$ critical point. The inset shows a plot of $S_2$ minus the leading term in Eq. (55), exposing the logarithmic dependence in $\ell$ of the subleading term.

tain universal information about the number of Goldstone modes, and the central charge of the underlying conformal field theory [47, 47, 48, 121, 127–131]. Extracting this information will require access to the large system sizes possible with PIGSFLI, which opens up the door for further exploration of entanglement properties and scaling in the ground states of bosonic lattice models.

Having explored the $\alpha = 2$ Rényi entanglement entropy, we now turn to the accessible entanglement entropy, named in this way due its original definition in terms of entanglement in a quantum many-body system accessible via local operations and classical communication [31]. In Fig. 12, we show results for a 1D Bose-Hubbard model of $N = 8$ bosons at unit-filling under an equal size bipartition. Similar results have already been reported in the literature [34, 36], and these should be considered as demonstrating the utility of PIGSFLI in computing this important quantity. Quantum Monte Carlo results (symbols) are shown as a function of projection length along with values computed via exact diagonalization (solid line) for the same interaction strengths and system sizes ($L = N = 8$) studied in Figs. 4 and 9. The accessible entanglement entropy $S_2^{\text{acc}}$ is bounded from above by the full Rényi entanglement $S_2$ and we find that it is considerably smaller, by a factor of 2 to 3 times in the Mott insulating phase and over 100 times in the superfluid phase. This is due to the fact that it is known to only be large near the quantum phase transition in this system [36] and goes to zero for the case of non-interacting bosons ($U/t \to 0$) where all entanglement is due to number fluctuations, or vanishes in the insulating phase, where the entanglement is non-accessible.

In Fig. 13, the symmetry-resolved entanglement entropy as a function of projection length is shown for the sector where the local particle number distribution $P_n$ is maximal, $n_{\text{max}}$, and it's two adjacent sectors, $n_{\text{max}-1}$ and $n_{\text{max}+1}$, with their corresponding exact diagonalization values shown as a solid horizontal line. Results are included for the same $L = N = 8$ system from Fig. 12 with interaction strengths $U/t = 0.5, 3.3, 10$. For reference, the local particle number distributions, $P_n$, are shown for each value of the interaction strength. For $U/t = 10$, due to

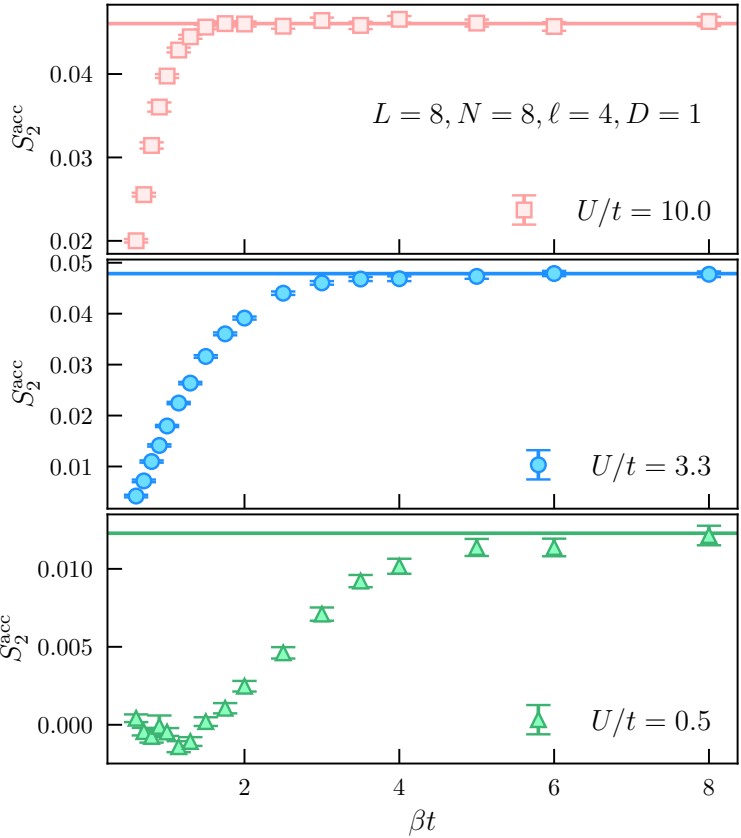

Figure 12: $\beta$-scaling of the accessible entanglement $S_2^{\text{acc}}$ in a 1D chain of $L = 8$ sites at unit-filling under equal bipartition of size $\ell = 4$. In contrast to the full Rényi entanglement entropies, larger values of $\beta$ are needed to achieve similar accuracy, especially in the superfluid phase. The solid horizontal lines denote the exact value of the accessible entanglement at the respective interaction strength.

the strong repulsion between particles, the ground state configuration tends to an insulating state with one boson per site at unit-filling. Since the subregion is of size $\ell = 4$, it is seen that $P_n$ is sharply peaked at a maximal value of $n_{\max} = 4$. In the superfluid phase with $U/t = 0.5$ there are considerably larger particle number fluctuations, resulting in a broader $P_n$, although with the peak still at $n_{\max} = 4$.

The symmetry-resolved entropies corresponding to $n_{\max}-1$ and $n_{\max}+1$ are equivalent, up to statistical fluctuations due to the chosen partition and the symmetries of the Bose-Hubbard model [36]. When particle fluctuations are large, the symmetry resolved entanglement is nearly identical for $n = n_{\max}, n_{\max} \pm 1$, whereas in the insulating phase, $n_{\max}$ may not correspond to maximal entanglement. For example, at $U/t = 10$, the contribution coming from the maximal sector is 10× smaller than the neighboring sectors, although their probability is much lower. This is because in the $n_{\max}$ sector, the ground state in the Mott insulating phase is $|\Psi_0\rangle_{\text{Mott}}^{n_{\max}} = |1, 1, \ldots, 1\rangle_A \otimes |1, 1, \ldots, 1\rangle_B$, which is unentangled. In the $n_{\max} - 1$ sector, the particles will once again try to repel each other, but there will be a vacancy in one of the $A$ sites. The ground state then becomes a superposition of the only two possible states that have a hole in the $A$ subregion that is adjacent to a doubly occupied site in the $B$ subregion: $|\Psi_0\rangle_{\text{Mott}}^{n_{\max}-1} = (|1, 1, \ldots, 0\rangle_A \otimes |2, 1, \ldots, 1\rangle_B + |0, 1, \ldots, 1\rangle_A \otimes |1, 1, \ldots, 2\rangle_B)/\sqrt{2}$, which can be shown to have a second Rényi entanglement entropy of $S_2 = \ln 2 \approx 0.69\ldots$ For the case of the insulating phase in the $n_{\max} + 1$ sector, an equivalent argument can be applied to understand the value of the entanglement entropy.

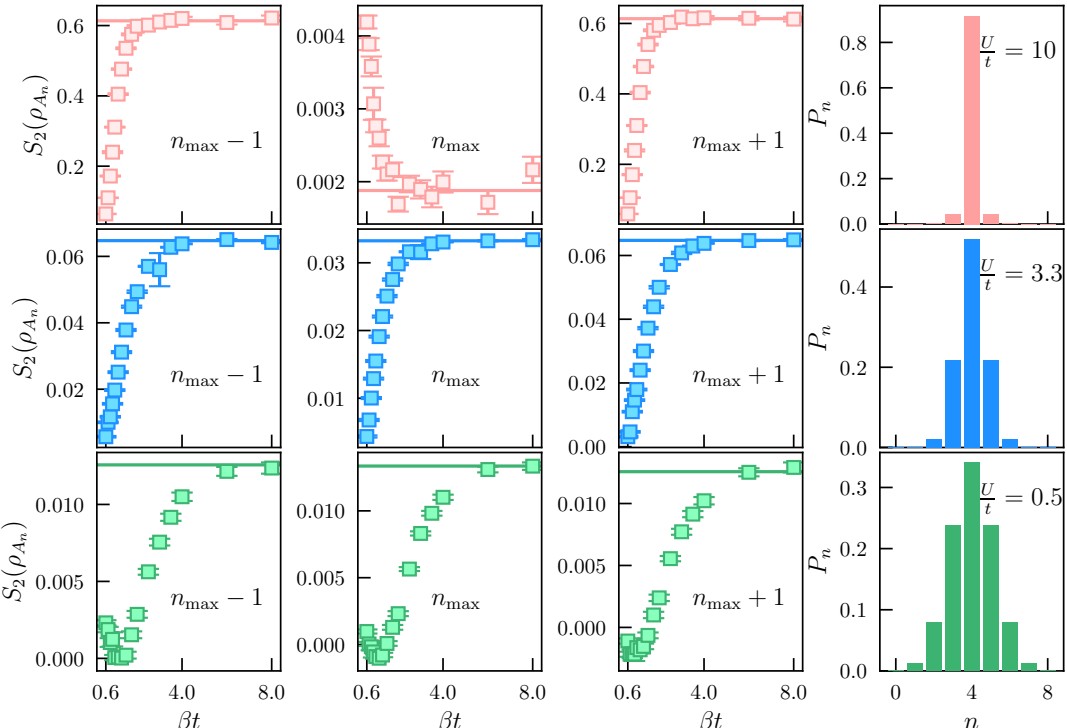

Figure 13: Imaginary time projection length $\beta$-scaling of the symmetry-resolved entanglement $S_2(\rho_{A_n})$ in a 1D Bose-Hubbard chain of $L = 8$ sites at unit-filling under an equal spatial bipartition of size $\ell = 4$. From top to bottom, the rows correspond to interaction strengths in the superfluid phase ($U/t = 0.5$), at the phase transition ($U/t = 3.3$), and in the Mott phase ($U/t = 10$). The first three columns correspond to entanglement entropies in the maximal local particle number sector $n_{max}$ (i.e., with the largest probability), and the two neighboring sectors $n_{max} - 1$ and $n_{max} + 1$. The solid horizontal lines are the exact diagonalization values for each symmetry resolved entanglement entropy. The rightmost column shows the local particle number distribution $P_n$ at each of the three interaction strengths.

As a practical matter, large $\beta$ values are required in Fig. 12 and Fig. 13 for the `PIGSFLI` algorithm to converge on results with small relative errors, however in all interaction regimes, we find agreement with the exact result. Improved sampling procedures can be directly implemented (e.g. performing parallel simulations in different restricted $n$-sectors) to improve efficiency and statistical convergence. These results represent the first quantum Monte Carlo measurement of the Rényi generalized accessible entanglement in the Bose-Hubbard model.

Fig. 14 shows results for the symmetry-resolved entanglement entropy as a function of local particle number sector $n$ for a 1D Bose-Hubbad model with $N = 64$ bosons at at unit filling under an equal spatial bipartition of size $\ell = 32$ near the quantum critical point, $U/t = 3.3$. This result demonstrates the capability of our algorithm to compute the symmetry-resolved entanglement entropy in much larger systems than have been previously possible, even in strongly interacting quantum many-body systems. The bottom panel of Fig. 14 shows the probability distribution of local particle number in the $A$ subregion. Notice that due to the onset of Mott insulating behavior, a similar behavior to Fig. 13 is observed, where the symmetry-resolved entanglement entropy at the maximal sector, $n_{max} = 32$, is actually smaller than it's two adjacent neighbors. In principle, the symmetry-resolved entanglement entropy is generally finite for the rest of the local particle numbers not shown, however as sectors outside of this range

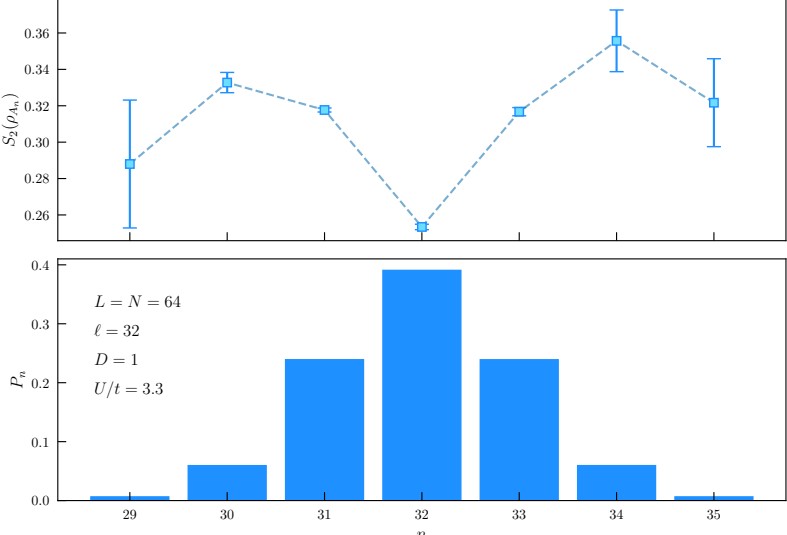

Figure 14: Symmetry-resolved Rényi entanglement entropies as a function of local particle number sector, $n$. The system is a one-dimensional lattice of $N = 64$ bosons at unit-filling under an equal spatial bipartition $\ell = 32$ near the critical interaction strength $U/t = 3.3$. The bottom panel shows the local particle number distribution, $P_n$, for this system.

occur with increasingly vanishing probability, in practice they cannot be sampled without large statistical errors. This is apparent in the figure as $|n - n_{max}| > 1$. The non-monotonic dependence of $S_2(\rho_{A_n})$ with the minimum occurring at $n_{max}$ can be interpreted as originating from the presence of holon and doublon quasiparticles when the subsystem is away from unit-filling, on the insulating side of the phase transition .

# 7 Conclusion

In this paper, we have introduced a ground state lattice Path Integral quantum Monte Carlo algorithm to compute the properties of interacting bosons at zero temperature. Our implementation, dubbed PIGSFLI, has been released as open source [30], and has been successfully tested by calculating the kinetic and potential energy of a one dimensional Bose-Hubbard chain, where we find perfect agreement up to stochastic uncertainty for small system sizes amenable to exact diagonalization. The algorithm was further expanded to allow for the calculation of the full, operationally accessible, and symmetry resolved spatial Rényi entanglement entropies where we again provided benchmarks against exact results across the phase diagram of the 1D Bose-Hubbard model. As has been previously reported, we observe that entanglement is sensitive to the quantum phase transition between the insulating and superfluid phases. To highlight the $O(L^D)$ performance of the quantum Monte Carlo implementation for a $D$-dimensional lattice of linear dimension $L$ we reported new results of spatial entanglement for a $D = 1$ chain of $L = 256$ sites at unit-filling. Moving beyond $D = 1$, we also demonstrated the entanglement scaling with boundary size for a $D = 2$ system of $M = 32 \times 32 = 1024$ lattice sites at unit-filling, with square subregions that ranged from as small as $\ell^D = 1 \times 1$ to $\ell^D = 20 \times 20$. These results are consistent with the entanglement boundary law, $S_2 \sim \ell$. Finally, by utilizing the superselection rule corresponding to fixed total particle number, we computed the accessible entanglement for a $D = 1$ Bose-Hubbard chain, as well as highlighting its value for a few symmetry resolved subsectors corresponding to $n$ particles in the subsystem near

the peak of the particle number distribution. While we have only studied these quantities in small systems, they demonstrate proof-of-principle calculations that can be straightforwardly extended to uncover the finite size scaling of this experimentally important entanglement measure.

There are many interesting avenues to pursue for further algorithmic developments including incorporating optimizations such as the "ratio method" [3,132] that have been previously utilized to improve sampling statistics in larger systems by building up the entanglement from a ratio of estimators for smaller spatial subregions. The addition of different lattice connectivities or moving to extended range Bose-Hubbard models (both hopping and interactions) presents no fundamental algorithmic challenge and will allow for the measurement of entanglement in the ground states of a large class of interacting lattice Hamiltonians. Moving to higher order Rényi entropies with $\alpha > 2$ is also possible by including additional replicas and the updates required to insert and remove SWAP kinks between them. Finally, it may also be possible to extend the algorithm to study entanglement measures more suitable for mixed states, such as the negativity.

## Acknowledgements

We thank N. Prokof'ev, N. S. Nichols, H. Barghathi, and C. Batista for fruitful discussions.

This work was supported in part by the NSF under Grant Nos. DMR-1553991 and DMR-2041995.

During manuscript preparation, ECD was supported by the Laboratory Directed Research and Development Early Career Research Funding of Los Alamos National Laboratory (LANL) under project number 20210662ECR. LANL is operated by Triad National Security, LLC, for the National Nuclear Security Administration of U.S. Department of Energy (Contract No. 89233218CNA000001).

## A   Obtaining and Running the Code

In the spirit of promoting open science, the PIGSFLI source code has been made public and can be obtained from the Del Maestro Group code repository [30] via:

```
git clone https://github.com/DelMaestroGroup/pigsfli.git
```

After compiling the code following the instructions in the repository [30], the executable pigsfli.e will be generated and simulations can now be performed. An example call would be:

```
./pigsfli.e -D 1 -L 4 -N 4 -l 2 -U 1.995 --mu 1.998 --beta 2.001
--seed 1968
```

In the call above, only some parameters have been set from the command-line, with the rest being set to their default values. The full list of parameters can be found on the code repository or can be seen by calling ./pigsfli.e --help .

In the first stage of the code, the chemical potential ($\mu$) is updated using the method in [86] until the average number of particles is at least within 33% of the target number $N$. To improve sampling efficiency, worldline updates are rejected if they would take the total particle number outside the interval $[N-1, N+1]$. In this stage, we also simultaneously perform a coarse tuning of the worm fugacity ($\eta$). We have chosen the coarse tuning to be $\eta \approx 1/\langle N_{\text{flats}}\rangle$, where $\langle N_{\text{flats}}\rangle$ is the average number of flat regions sampled each time that a distribution $P(N)$ was built.

A fine tuning of $\eta$ is then performed until the fraction of worldline configurations with no worm ends, or diagonal fraction (`Z-frac` in the code output shown) is within a desired window. This fraction needs to be low enough that physical observables can be measured, but high enough that worm ends are present to help push the dynamics of the worldline configuration forward. We've chosen a window between 40% and 45% for the target value of the diagonal fraction. The $\eta$ updating is performed by asymmetrically increasing or decreasing its value by some multiplicative constant. For example, if the fraction is too low, we decrease $\eta$ by 50% and if too high, increase it by 45%. Note that we still keep updating $\mu$ in this stage in case that the new $\eta$ causes the average total particle number to stray too far away from the target value.

After tuning $\mu$ and $\eta$, there is an equilibration stage where worldline updates are performed for a chosen number of Monte Carlo sweeps, but no measurements are performed. The number of equilibration sweeps is set to a default value but can also be set from the command line.

And finally, the last stage is the main Monte Carlo loop, where not only worldline updates are performed, but also measurements of the desired quantities. After the main loop, the number of times that each update was accepted over the number of times it was proposed is shown. Notice that the "forward" and "backward" updates of each pair of complementary updates is accepted roughly the same time, as expected from the principle of detailed balance.

The code output for the example call above is:

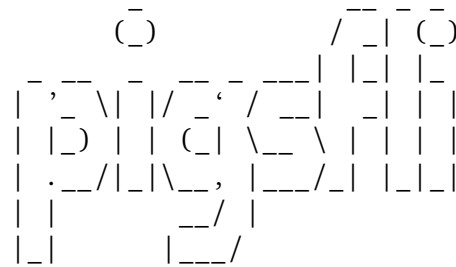

```
Path-Integral Ground State (Monte Carlo) For Lattice
  Implementations

Stage (1/3): Determining mu and eta...

mu: 1.998 eta: 0.5 Z-frac: 13.004%
N      P(N)
4      **
5      *****************************************************
<N>: 4.98985

mu: -0.291113 eta: 0.153571 Z-frac: 18.221%
N      P(N)
3      ***********
4      *****************************************************
5      ****************************
<N>: 4.16743

Fine tuning eta... (Want: 10% < Z-frac < 15%)

mu: -0.291113 eta: 0.16197 Z-frac: 20.453%
```

```
N       P(N)
3       ***
4       ***********************************************
5       *********************************************
<N>: 4.4351

mu: −1.02474 eta: 0.0809852 Z−frac: 29.8528%
N       P(N)
4       *****************************************************
5       ************************
<N>: 4.24996

mu: −1.02474 eta: 0.0404926 Z−frac: 54.4598%
N       P(N)
3       *********************
4       *****************************************************
5       ****************
<N>: 3.94856

mu: −1.02474 eta: 0.0587143 Z−frac: 40.3%
N       P(N)
3       *****
4       ******************************************************
5       *************************
<N>: 4.2089

Stage (2/3): Equilibrating...

Stage (3/3): Main Monte Carlo loop...

————————  Detailed  Balance  ————————

Insert Worm: 38470/2772581
Delete Worm: 36544/44720

Insert Anti: 37646/2090863
Delete Anti: 35628/39821

InsertZero Worm: 396535/6277132
DeleteZero Worm: 398573/475282

InsertZero Anti: 433392/4407244
DeleteZero Anti: 435430/522308

InsertBeta Worm: 396510/6280347
DeleteBeta Worm: 398416/476129

InsertBeta Anti: 432507/4409839
DeleteBeta Anti: 434413/519043
```

```
Advance Head: 1572275/1578564
Recede  Head: 1572292/1597065

Advance Tail: 1573978/1598309
Recede  Tail: 1575285/1581400

IKBH: 1508373/3785551
DKBH: 1509684/1697842

IKAH: 946308/3783103
DKAH: 945886/1120626

IKBT: 945858/2638223
DKBT: 945478/1119408

IKAT: 1508438/3787390
DKAT: 1507908/1697276

SWAP: 5932134/14723038
UNSWAP: 5932132/14811285

SWAP Advance Head: 313417/321052
SWAP Recede Head: 313426/344143

SWAP Advance Tail: 312995/342388
SWAP Recede Tail: 312737/319738

Elapsed time: 13.5167 seconds
```

In Stage (1/3), histograms of the total particle number distribution are shown, where each of the asterisks (*) represents a normalized count. For canonical ensemble simulations, like the one shown above, the only particle numbers visited are $N - 1$, $N$, and $N + 1$, where $N$ is the target number of particles. A grand canonical simulation will show histograms with more particle numbers than these. Once the peak of the distribution is at $N$, and it's at least 33

Stage (2/3), the code is ran without taking any measurement as a en equilibration step. The number of equilibration steps are currently determined by the sweeps parameter from the command line.

Finally, Stage (3/3) is where measurements are performed and samples collected. Once the desired number of samples are collected, the simulation stops. Near the bottom of the terminal output above, the number of times that each of the updates is accepted and proposed are shown as a fraction. The number of times that the update is accepted is shown in the numerator, whereas the times that it was proposed is shown in the denominator.

The total run time of equilibration and Main Monte Carlo loops is shown at the bottom of the output, in seconds.

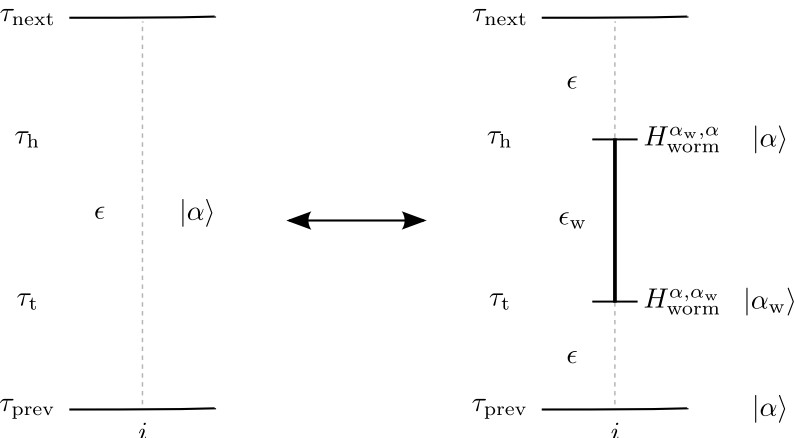

Figure 15: **Insert/Delete worm.** A worm is inserted by first randomly sampling imaginary times $\tau_t, \tau_h$ inside an also randomly sampled flat interval, delimited by the times $\tau_{prev}$ and $\tau_{next}$. A worm head and tail are then inserted at $\tau_h$ and $\tau_t$, respectively. $\epsilon$ and $\epsilon_w$ correspond to the eigenvalues of the diagonal part of the Hamiltonian for Fock states $|\alpha\rangle$ and $|\alpha_w\rangle$, respectively. The case of $\tau_h < \tau_t$ is also valid and would correspond to an antiworm insertion, in which case a particle is first deleted by the worm head, then one created by the tail.

# B  Original Finite Temperature Updates

## B.1  Insert/Delete worm (or antiworm)

**Insert**: The insert worm update creates a finite particle worldline on a single site through the insertion of a creation operator at time $\tau_t$ and destruction operator at $\tau_h$ within a *flat* region of imaginary time delimited by imaginary times $\tau_{prev}$ and $\tau_{next}$ where the particle number does not change. A worm has a tail located at $\tau_t$ and head at $\tau_h$ and these times are subject to the constraint: $\tau_h > \tau_t$. This update is proposed only if there are no worm ends already present in the configuration.

An antiworm can instead be inserted if the sampled time of the worm head is smaller than the tail: $\tau_h < \tau_t$. This will first annihilate a particle and create one at a later time inside the flat region. This is only possible if there is at least one particle in the chosen flat region. Thus, if the sampled times correspond to an antiworm and there are no particles in the flat region, the update is rejected. For simplicity, we'll refer to either of these two types of path discontinuities as a worm when describing the updates. The insert worm update is illustrated in Fig. 15 and proceeds as follows:

0. Attempt update with probability $p_{insert}$. This is the bare probability of attempting this move from amongst the entire pool of updates. Every other update will have a similar bare attempt probability.

1. Randomly sample a flat region with probability $1/N_{flats}$, where $N_{flats}$ is the total number of flat regions.

2. Randomly sample the worm head time $\tau_h \in [\tau_{prev}, \tau_{next})$ with probability $1/(\tau_{next} - \tau_{prev})$.

3. Randomly sample the worm tail time $\tau_t \in [\tau_{prev}, \tau_{next})$ with probability $1/(\tau_{next} - \tau_{prev})$. Reject update if $\tau_h < \tau_t$ and there are no particles in the flat interval.

4. Calculate the diagonal energy difference $\Delta V \equiv \epsilon_{\mathrm{w}} - \epsilon$, where $\epsilon_{\mathrm{w}}$ is the diagonal energy of the segment of path inside the flat region with more particles, and $\epsilon$ the diagonal energy in the segment with less particles.

5. Sample a random and uniformly distributed number $r \in [0, 1)$.

6. Check,
   If $r < R_{\mathrm{insert}}$, insert worm into worldline configuration.
   Else, reject update and leave worldlines unchanged.

**Delete**: The complementary update of a worm insertion is a worm deletion, and it proceeds as follows:

0. Attempt update with probability $p_{\mathrm{delete}}$.

1. Calculate diagonal energy difference $\Delta V \equiv \epsilon_{\mathrm{w}} - \epsilon$.

2. Sample a random and uniformly distributed number $r \in [0, 1)$.

3. Check,
   If $r < R_{\mathrm{delete}}$, delete worm from worldline configuration.
   Else, reject update and leave worldlines unchanged.

The constants $R_{\mathrm{insert}}$ and $R_{\mathrm{delete}}$ are the Metropolis acceptance ratios for the complementary pair of insert worm and delete worm updates and are computed using Eq. (33). Evaluating the configurational weights after and before worm insertion according to Eq. (31) and taking their ratio gives:

$$\frac{W_{\mathrm{insert}}}{W_{\mathrm{delete}}} = \eta^2 n_{\mathrm{w}} e^{-(\epsilon_{\mathrm{w}} - \epsilon)(\tau_{\mathrm{h}} - \tau_{\mathrm{t}})} \,. \tag{B.1}$$

Here, $n_{\mathrm{w}}$ is the number of particles in the path segment with the extra particle (denoted by the subscript w), or the segment after the worm tail, and $\epsilon_{\mathrm{w}}$ is the diagonal energy also in this path segment.

The proposal probabilities are obtained by multiplying all the probabilities associated with each step of the update's decision process:

$$\frac{P_{\mathrm{delete}}}{P_{\mathrm{insert}}} = \frac{p_{\mathrm{delete}}}{p_{\mathrm{insert}}} (\tau_{\mathrm{next}} - \tau_{\mathrm{prev}})^2 N_{\mathrm{flats}} \,. \tag{B.2}$$

The total acceptance ratio for the worm insertion update thus becomes:

$$R_{\mathrm{insert}} = \eta^2 n_{\mathrm{w}} e^{-(\epsilon_{\mathrm{w}} - \epsilon)(\tau_{\mathrm{h}} - \tau_{\mathrm{t}})} \frac{p_{\mathrm{delete}}}{p_{\mathrm{insert}}} (\tau_{\mathrm{next}} - \tau_{\mathrm{prev}})^2 N_{\mathrm{flats}} \tag{B.3}$$

and for worm deletion, we have the reciprocal:

$$R_{\mathrm{delete}} = \frac{1}{R_{\mathrm{insert}}} \,. \tag{B.4}$$

### B.2 Advance/Recede

This update update can be proposed when there is at least one worm end (head/tail) present. A worm end is selected randomly, then it is moved backward or forward in the imaginary time direction. The update is illustrated in Fig. 16 and it proceeds as follows:

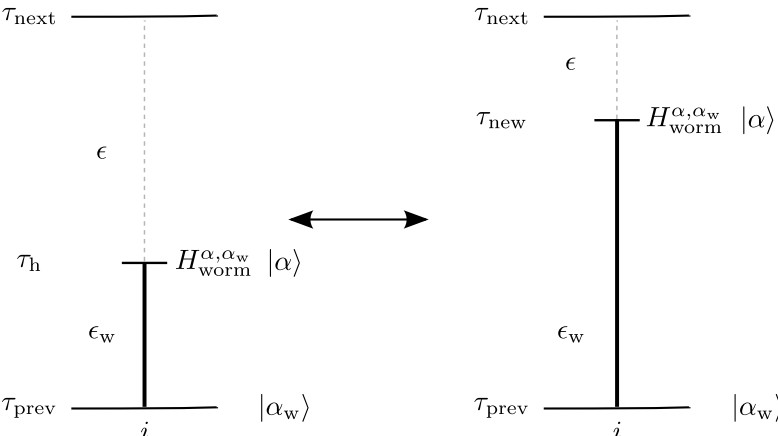

Figure 16: **Advance/Recede.** Worm end is selected at random. It is then moved to a randomly sampled time inside the flat interval. The diagram above illustrates the example of advancing/receding a head in imaginary time. Either the worm head or tail can be timeshifted.

0. Randomly choose to move worm head or tail. If there is only one end, choose that one.

1. Depending on the worm end selected, sample a new worm end time $\tau_{\mathrm{new}} \in [\tau_{\mathrm{prev}}, \tau_{\mathrm{next}})$ from a truncated exponential distribution

$$p(\tau_{\mathrm{new}}) = \begin{cases} \frac{\epsilon_{\mathrm{w}} - \epsilon}{1 - e^{-(\epsilon_{\mathrm{w}} - \epsilon)(\tau_{\mathrm{next}} - \tau_{\mathrm{prev}})}} e^{-(\epsilon_{\mathrm{w}} - \epsilon)(\tau_{\mathrm{new}} - \tau_{\mathrm{prev}})} & \text{head}, \\ \frac{\epsilon - \epsilon_{\mathrm{w}}}{1 - e^{-(\epsilon - \epsilon_{\mathrm{w}})(\tau_{\mathrm{next}} - \tau_{\mathrm{prev}})}} e^{-(\epsilon - \epsilon_{\mathrm{w}})(\tau_{\mathrm{new}} - \tau_{\mathrm{prev}})} & \text{tail}. \end{cases} \tag{B.5}$$

The reason to sample the new time of the chosen worm end from a truncated exponential distributions is that the Metropolis acceptance ratio becomes unity and the update is always accepted. This is because the ratio of weights for the advance/recede update, as computed from Eq. (27), is either:

$$\frac{W_{\mathrm{advance}}}{W_{\mathrm{recede}}} = e^{-(\epsilon_{\mathrm{w}} - \epsilon)(\tau_{\mathrm{new}} - \tau_{\mathrm{h}})}, \tag{B.6}$$

for advance/recede of a worm head and:

$$\frac{W_{\mathrm{advance}}}{W_{\mathrm{recede}}} = e^{(\epsilon_{\mathrm{w}} - \epsilon)(\tau_{\mathrm{new}} - \tau_{\mathrm{t}})}, \tag{B.7}$$

for advance/recede of a worm tail. As for the proposal probabilities, they are computed from Eq. (B.5). Taking the ratio of the truncated exponential distribution $p(\tau)$ between the new and old imaginary times, we obtain a complete cancellation and acceptance ratio for this update is just unity:

$$R_{\mathrm{advance}} = R_{\mathrm{recede}} = 1. \tag{B.8}$$

### B.3 Insert/Delete kink before worm head

Worm ends can move to neighboring lattice sites by inserting a kink either before or after them. The imaginary time interval at which the kink can be inserted is delimited from above by the worm head itself and from below by the largest of the two lower bounds of the relevant flat regions on both sites. We reject all updates that would interfere with kinks on adjacent sites. The update is illustrated in Fig. 17, for the case of a worm head. In the figure, the matrix

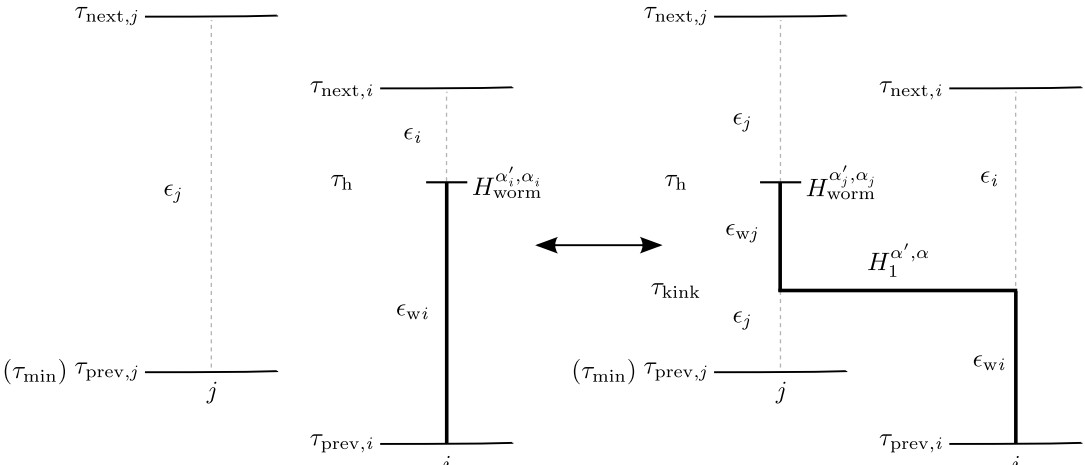

Figure 17: **Insert/Delete kink before head.** A kink is inserted between neighboring sites $i,j$ before the worm head at a randomly sampled time $\tau_{\text{kink}} \in [\tau_{\min}, \tau_h)$ and the head moved to the site where the particle hops to. The lower bound of the sampling interval $\tau_{\min}$ is chosen to be the largest out of the two lower bounds of the corresponding flat intervals. This ensures that the new kink does not interfere with other kinks, which simplifies the implementation. The complementary update is to delete the kink and move the head back to the site from where the particle is hopping from.

element corresponding to the worm end, in this case a head, is $H_{\text{worm}}^{\alpha'_i, \alpha_i} = \left\langle \alpha'_i \right| H_{\text{worm}} \left| \alpha_i \right\rangle$, where $\left| \alpha_i \right\rangle$ and $\left| \alpha'_i \right\rangle$ are the Fock states preceding and following the worm head, when it is on site $i$ (i.e, before the kink is inserted). The matrix element $H_{\text{worm}}^{\alpha'_j, \alpha_j} = \left\langle \alpha'_j \right| H_{\text{worm}} \left| \alpha_j \right\rangle$ is defined analogously, but for when the worm head is on site $j$ (i.e, after the kink is inserted). The update proceeds as follows:

**Insert kink before head:**

0. Attempt update with probability $p_{\text{insertKinkBeforeHead}}$.

1. Randomly sample a nearest neighbor $j$ of site $i$ where the head resides with probability $1/N_{\text{nn}}$, where $N_{\text{nn}}$ is the number of nearest-neighbor sites.

2. Randomly sample an imaginary time $\tau_{\text{kink}} \in [\tau_{\min}, \tau_h)$ with probability $1/(\tau_h - \tau_{\min})$.

3. Compute the diagonal energy differences $\Delta V_i = \epsilon_{\text{w}i} - \epsilon$ and $\Delta V_i = \epsilon_{\text{w}j} - \epsilon$, where the $i, j$ subscripts denote the source and destination sites of the kink.

4. Sample a random and uniformly distributed number $r \in [0, 1)$.

5. Check,
   If $r < R_{\text{insertKinkBeforeHead}}$, insert kink into worldline configuration.
   Else, reject update and leave worldlines unchanged.

**Delete kink before head:**

0. Attempt update with probability $p_{\text{deleteKinkBeforeHead}}$.

1. Compute the diagonal energy differences $\Delta V_i = \epsilon_{\text{w}i} - \epsilon$ and $\Delta V_i = \epsilon_{\text{w}j} - \epsilon$.

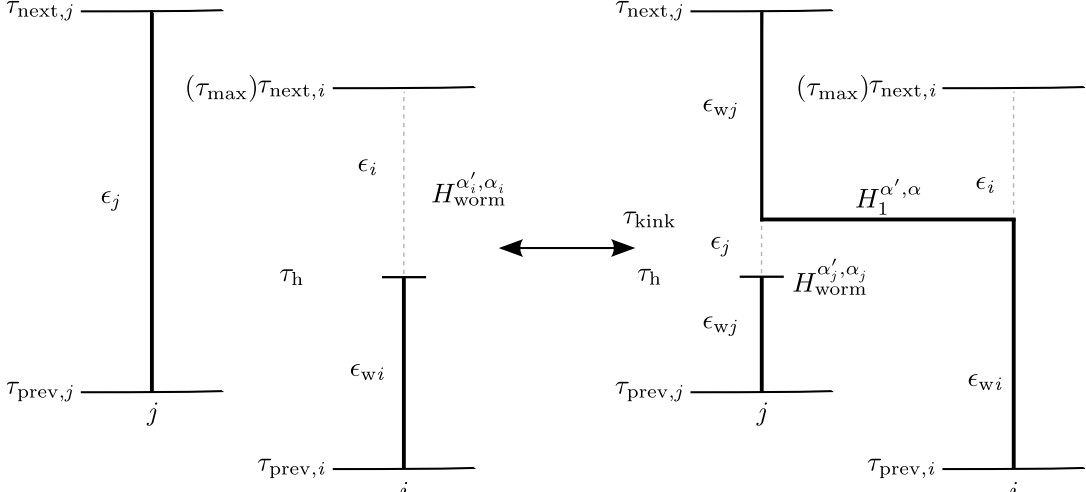

Figure 18: **Insert/Delete kink after head.** A kink is inserted between neighboring sites $i,j$ after the worm head at a randomly sampled time $\tau_{\mathrm{kink}} \in [\tau_{\mathrm{h}}, \tau_{\mathrm{max}})$ and the head moved to the site where the particle hops to. The upper bound of the sampling interval $\tau_{\mathrm{max}}$ is chosen to be the smallest out of the two upper bounds of the corresponding flat intervals. This ensures that the new kink does not interfere with other kinks, which simplifies the implementation. The complementary update is to delete the kink and move the head back to the site from where the particle is hopping from.

2. Sample a random and uniformly distributed number $r \in [0,1)$.

3. Check,
   If $r < R_{\mathrm{deleteKinkBeforeHead}}$, delete kink from worldline configuration.
   Else, reject update and leave worldlines unchanged.

The Metropolis acceptance ratio for inserting a kink before a worm head becomes:

$$R_{\mathrm{insertKinkBeforeHead}} = t\,n_{\mathrm{w}j}\,e^{(\Delta V_i - \Delta V_j)(\tau_{\mathrm{h}} - \tau_{\mathrm{kink}})}\frac{p_{\mathrm{deleteKinkBeforeHead}}}{p_{\mathrm{insertKinkBeforeHead}}}(\tau_{\mathrm{h}} - \tau_{\mathrm{min}})N_{\mathrm{nn}}\,, \qquad (B.9)$$

where $n_{\mathrm{w}j}$ is the number of particles in the segment of site $j$ with the extra particle. $\tau_{\mathrm{min}}$ is the lower bound of the sampling interval, defined such that the update does not interfere with other kinks. Out of the possible candidates for the lower bound in sites $i, j$ it is the largest time: $\tau_{\mathrm{min}} = \max\{\tau_{\mathrm{prev},i}, \tau_{\mathrm{prev},j}\}$. Other variations of this update, in which kinks can be inserted after a head, before a tail, and after a tail, will adapt the notational conventions introduced here. For the complementary update of deleting the kink before the head, the Metropolis acceptance ratio is just the reciprocal:

$$R_{\mathrm{deleteKinkBeforeHead}} = \frac{1}{R_{\mathrm{insertKinkBeforeHead}}}. \qquad (B.10)$$

### B.4 Insert/Delete kink after worm head

A kink can also be inserted or deleted after the worm head. The update is illustrated in Fig. 18. The randomly sampled imaginary time $\tau_{\mathrm{kink}}$ is now in the interval $[\tau_{\mathrm{h}}, \tau_{\mathrm{max}})$. The maximum time is chosen as the smallest of the upper bounds on both sites, such that the inserted kink does not interfere with other kinks: $\tau_{\mathrm{max}} = \min\{\tau_{\mathrm{next},i}, \tau_{\mathrm{next},j}\}$. Other than the kink now being at a

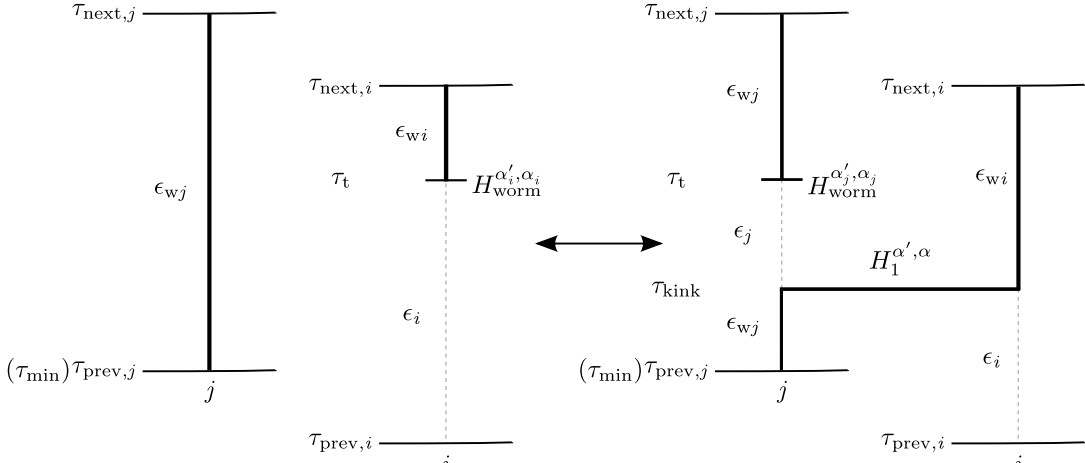

Figure 19: **Insert/Delete kink before tail.** A kink is inserted between neighboring sites $i,j$ before the worm tail at a randomly sampled time $\tau_{\mathrm{kink}} \in [\tau_{\min}, \tau_{\mathrm{t}})$ and the tail moved to the site where the particle hops from. The lower bound of the sampling interval $\tau_{\min}$ is chosen to be the largest out of the two lower bounds of the corresponding flat intervals. This ensures that the new kink does not interfere with other kinks, which simplifies the implementation. The complementary update is to delete the kink and move the tail back to the site from where the particle is hopping to.

later imaginary time than where the head is, the procedure is analogous to inserting/deleting a kink before head. The Metropolis acceptance ratio for insertion of a kink after the head is:

$$R_{\mathrm{insertKinkAfterHead}} = t n_{\mathrm{w}j} e^{(-\Delta V_i + \Delta V_j)(\tau_{\mathrm{kink}} - \tau_{\mathrm{h}})} \frac{p_{\mathrm{deleteKinkAfterHead}}}{p_{\mathrm{insertKinkAfterHead}}} (\tau_{\max} - \tau_{\mathrm{h}}) N_{\mathrm{nn}} , \qquad \text{(B.11)}$$

and for deletion of a kink after the head:

$$R_{\mathrm{deleteKinkAfterHead}} = \frac{1}{R_{\mathrm{deleteKinkAfterHead}}} . \qquad \text{(B.12)}$$

### B.5 Insert/Delete kink before worm tail

The update is illustrated in Fig. 19. The imaginary time of the kink is now sampled such that $\tau_{\mathrm{kink}} \in [\tau_{\min}, \tau_{\mathrm{t}})$. The Metropolis acceptance ratio is:

$$R_{\mathrm{insertKinkBeforeTail}} = t n_{\mathrm{w}j} e^{(-\Delta V_i + \Delta V_j)(\tau_{\mathrm{t}} - \tau_{\mathrm{kink}})} \frac{p_{\mathrm{insertKinkBeforeTail}}}{p_{\mathrm{deleteKinkBeforeTail}}} (\tau_{\mathrm{t}} - \tau_{\min}) N_{\mathrm{nn}}. \qquad \text{(B.13)}$$

### B.6 Insert/Delete kink after worm tail

The update is illustrated in Fig. 20. The imaginary time of the kink is now sampled such that $\tau_{\mathrm{kink}} \in [\tau_{\mathrm{t}}, \tau_{\max})$. The Metropolis acceptance ratio is:

$$R_{\mathrm{insertKinkAfterTail}} = t n_{\mathrm{w}j} e^{(\Delta V_i - \Delta V_j)(\tau_{\mathrm{kink}} - \tau_{\mathrm{t}})} \frac{p_{\mathrm{insertKinkAfterTail}}}{p_{\mathrm{deleteKinkAfterTail}}} (\tau_{\max} - \tau_{\mathrm{t}}) N_{\mathrm{nn}}. \qquad \text{(B.14)}$$

The updates proposed in this appendix satisfy ergodicity at finite temperature, however, the set of additional updates shown in Section 3.5 are required to satisfy ergodicity at $T = 0$.

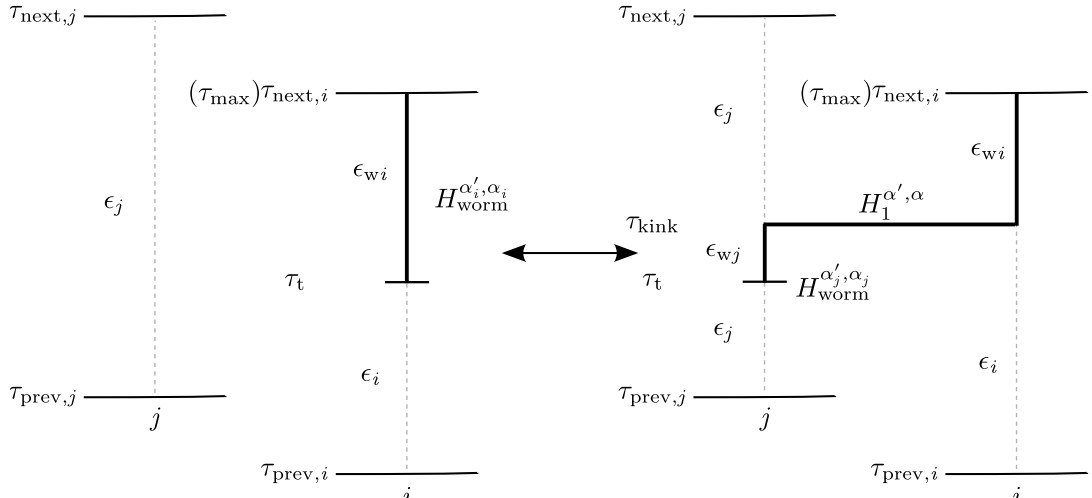

Figure 20: **Insert/Delete kink after tail.** A kink is inserted between neighboring sites $i,j$ after the worm tail at a randomly sampled time $\tau_{\text{kink}} \in [\tau_{\text{t}}, \tau_{\text{max}})$ and the tail moved to the site where the particle hops from. The upper bound of the sampling interval $\tau_{\text{max}}$ is chosen to be the smallest out of the two upper bounds of the corresponding flat intervals. This ensures that the new kink does not interfere with other kinks, which simplifies the implementation. The complementary update is to delete the kink and move the tail back to the site from where the particle is hopping to.

## C  Potential Energy

The ground state expectation value of the potential energy can be obtained via:

$$\langle H_0(\tau) \rangle = \langle \psi_0 | H_0(\tau) | \psi_0 \rangle. \tag{C.1}$$

Rewriting the ground state in terms of the trial wavefunction and inserting various resolutions of the identity operator like it was done for the kinetic energy:

$$\langle H_0(\tau) \rangle = \lim_{\beta \to \infty} \sum_{\alpha_\beta, \alpha'', \alpha', \alpha_0} C^*_{\alpha_\beta} C_{\alpha_0} \langle \alpha_\beta | e^{-\beta/2} | \alpha'' \rangle \langle \alpha'' | H_0(\tau) | \alpha' \rangle \langle \alpha' | e^{-\beta/2} | \alpha_0 \rangle, \tag{C.2}$$

where $H_0(\tau) = e^{\tau H} H_0 e^{-\tau H}$. Recall that $H_0$ is also the diagonal part of the Hamiltonian, such that $H_0 | \alpha \rangle = \epsilon_\alpha$, where $\epsilon_\alpha$ is the potential energy of state $\alpha$, and the specific form of these matrix elements is given in Eq. (23). Acting with $H_0(\tau)$ on the state $\alpha'$, a Kronecker-delta function is picked up, which will get rid of the $\alpha''$ summation:

$$\langle H_0(\tau) \rangle = \lim_{\beta \to \infty} \sum_{\alpha_\beta, \alpha', \alpha_0} \epsilon_{\alpha'}(\tau) C^*_{\alpha_\beta} C_{\alpha_0} \langle \alpha_\beta | e^{-\beta/2} | \alpha' \rangle \langle \alpha' | e^{-\beta/2} | \alpha_0 \rangle. \tag{C.3}$$

Due to the convolutional property of propagators:

$$\sum_{\alpha'} \langle \alpha_\beta | e^{-\beta/2} | \alpha' \rangle \langle \alpha' | e^{-\beta/2} | \alpha_0 \rangle = \rho(\alpha_\beta, \alpha_0; \beta). \tag{C.4}$$

Using then the definition of the propagator in Eq. (17) and the configurational weight in Eq. (27), it is seen that:

$$\langle H_0(\tau) \rangle = \lim_{\beta \to \infty} \sum_{Q, \boldsymbol{\alpha}_Q} \int d\boldsymbol{\tau}_Q \, \epsilon_{\alpha'}(\tau) W_0(Q, \boldsymbol{\alpha}_Q, \boldsymbol{\tau}_Q) = \lim_{\beta \to \infty} \langle \epsilon_{\alpha'}(\tau) \rangle. \tag{C.5}$$

At the moment, the potential energy is only at a time $\tau$ away from $\beta/2$. Averaging over a time window of size $\Delta\tau$ centered around $\beta/2$, the potential energy gives:

$$\langle H_0\rangle = \frac{1}{\Delta\tau}\lim_{\beta\to\infty}\int_{-\Delta\tau/2}^{\Delta\tau/2}d\tau\langle\epsilon_{\alpha'}(\tau)\rangle = \frac{1}{\Delta\tau}\lim_{\beta\to\infty}\int_{\beta/2-\Delta\tau/2}^{\beta/2+\Delta\tau/2}d\tau\langle\epsilon_{\alpha'}(\tau)\rangle. \qquad\text{(C.6)}$$

where the transformation $\tau \to \beta/2 + \tau$ was used. Therefore, the estimator for the average ground state potential energy is:

$$\langle H_0\rangle_{\mathrm{MC}} \approx \frac{1}{\Delta\tau}\int_{\beta/2-\Delta\tau/2}^{\beta/2+\Delta\tau/2}d\tau\langle\epsilon_{\alpha'}(\tau)\rangle_{\mathrm{MC}}, \qquad\text{(C.7)}$$

where $\epsilon_{\alpha'}(\tau) = \frac{U}{2}\sum_i n_i^{\alpha'}(\tau)(n_i^{\alpha'}(\tau)-1)-\mu\sum_i n_i^{\alpha'}(\tau)$. The operator $n_i(\tau)$ counts the number of particles at site $i$ and imaginary time $\tau$, for a Fock state $\alpha'$, which can be computed directly in the simulation. The potential energy estimator becomes exact in the limit $\beta \to \infty$.

## D   Kinetic Energy

In the Heisenberg representation, the kinetic energy operator, which is also just the off-diagonal term of the Bose-Hubbard Hamiltonian in Eq. (19), is:

$$H_1(\tau) = e^{\tau H}H_1 e^{-\tau H}, \qquad\text{(D.1)}$$

where $\tau$ is an imaginary time. The ground state expectation value is, in principle, obtained in the usual way, by sandwiching $H_1(\tau)$ in between the ground state $|\psi_0\rangle$:

$$\langle H_1(\tau)\rangle = \langle\psi_0|H_1(\tau)|\psi_0\rangle. \qquad\text{(D.2)}$$

In practice, the expression above cannot be computed directly since the ground state is not exactly known. Instead, an estimator is needed that can be computed using our algorithm. Recall the projection relation: $|\psi_0\rangle = \lim_{\beta\to\infty}e^{-\frac{\beta}{2}H}|\psi_T\rangle$, where $|\psi_T\rangle$ is a trial wavefunction. Substituting this projection, the expectation value becomes:

$$\langle H_1(\tau)\rangle = \lim_{\beta\to\infty}\langle\psi_T|e^{-(\frac{\beta}{2}-\tau)H}H_1 e^{-(\frac{\beta}{2}+\tau)H}|\psi_T\rangle. \qquad\text{(D.3)}$$

In general, for a path-integral Monte Carlo algorithm, it is desirable to first write the estimator in terms of imaginary-time propagators. To do this, rewrite the expectation value in terms of the trial wavefunction expansion in the Fock basis: $|\psi_T\rangle = \sum_\alpha C_\alpha|\alpha\rangle$ and then insert some resolutions of the identity:

$$\langle H_1(\tau)\rangle = \lim_{\beta\to\infty}\sum_{\alpha_\beta\alpha''\alpha'\alpha_0}\langle\psi_T|\alpha_\beta\rangle\langle\alpha_\beta|e^{-(\frac{\beta}{2}-\tau)H}|\alpha''\rangle\langle\alpha''|H_1|\alpha'\rangle\langle\alpha'|e^{-(\frac{\beta}{2}+\tau)H}|\alpha_0\rangle\langle\alpha_0|\psi_T\rangle. \quad\text{(D.4)}$$

Rewriting the wavefunction coefficients as $\langle\psi_T|\alpha_\beta\rangle = C_{\alpha_\beta}^*$ and $\langle\alpha_0|\psi_T\rangle = C_{\alpha_0}$, the kinetic energy matrix elements as $H_1^{\alpha'',\alpha'} = \langle\alpha''|H_1|\alpha'\rangle$, and the short-time imaginary propagator between two states over an interval $\tau$ as: $\rho(\alpha,\alpha';\tau) = \langle\alpha|e^{-\tau H}|\alpha'\rangle = \langle\alpha'|e^{-\tau H}|\alpha\rangle$, the ground state expectation value of the kinetic energy becomes:

$$\langle H_1(\tau)\rangle = \lim_{\beta\to\infty}\sum_{\alpha_0\alpha''\alpha'\alpha_\beta}C_{\alpha_\beta}^* C_{\alpha_0}\rho(\alpha_\beta,\alpha'';\frac{\beta}{2}-\tau)H_1^{\alpha''\alpha'}\rho(\alpha',\alpha_0;\frac{\beta}{2}+\tau). \qquad\text{(D.5)}$$

Using Eq. (17), each of the propagators in the equation above can be expanded in the number of kinks inside the time interval they span (i.e, $[\beta/2 + \tau, \beta]$ for the leftmost propagator and $[0, \beta/2 + \tau]$ for the rightmost one).

In Eq. (D.5), the rightmost propagator will propagate the state $\alpha_0$ from $\tau = 0$ to state $\alpha'$, at $\tau = \beta/2 + \tau$, the time of the matrix element $H_1^{\alpha'', \alpha'}$. The leftmost propagator can be thought of as propagating in the direction of decreasing time, taking the state $\alpha_\beta$ at $\tau = \beta$ to state $\alpha''$, at $\tau = \beta/2 + \tau$, which is an interval of size $\beta/2 - \tau$. If the two states $\alpha''$ and $\alpha'$ differ only by a particle hop from one site to an adjacent site, the matrix element $H_1^{\alpha'', \alpha'}$ is just a kink connecting the states, and zero otherwise, according to the kinetic matrix element in Eq. (24). In Eq. (D.5), take the example of expanding the leftmost and rightmost propagators up to orders $Q_-$ and $Q_+$, respectively. The propagator on the right gives:

$$\rho\left(\alpha_\beta, \alpha''; \frac{\beta}{2} + \tau\right) = \sum_{\alpha_{+_1} \dots \alpha_{Q_+ - 1}} \int_0^{\tau_{Q_+ + 1} = \beta} d\tau_{Q_+} \int_0^{\tau_{Q_+}} d\tau_{Q_+ - 1} \cdots \int_0^{\tau_{2_+}} d\tau_{1_+}$$
$$\times \left[ (-1)^{Q_+} e^{-\epsilon_{\alpha_{0_+}} \tau_{1_+}} \prod_{q_+ = 1}^{Q_+} e^{-\epsilon_{\alpha_{q_+}} (\tau_{q_+ + 1} - \tau_{q_+})} H_1^{\alpha_{q_+}, \alpha_{q_+ - 1}} \right]. \quad \text{(D.6)}$$

The propagator on the left gives:

$$\rho\left(\alpha', \alpha_0; \frac{\beta}{2} - \tau\right) = \sum_{\alpha_{-_1} \dots \alpha_{Q_- - 1}} \int_0^{\tau_{Q_- + 1} = \frac{\beta}{2} + \tau} d\tau_{Q_-} \int_0^{\tau_{Q_-}} d\tau_{Q_- - 1} \cdots \int_0^{\tau_{2_-}} d\tau_{1_-}$$
$$\times \left[ (-1)^{Q_-} e^{-\epsilon_{\alpha_{0_-}} \tau_{1_-}} \prod_{q_- = 1}^{Q_-} e^{-\epsilon_{\alpha_{q_-}} (\tau_{q_- + 1} - \tau_{q_-})} H_1^{\alpha_{q_-}, \alpha_{q_- - 1}} \right]. \quad \text{(D.7)}$$

The subscripts $-, +$ can be dropped in favor of an "absolute" enumeration of the imaginary times, which will be relative to the total number of kinks $Q$: $\tau_{1_+} \to \tau_1, \tau_{2_+} \to \tau_2, \dots, \tau_{Q_-} \to \tau_Q$. Under this new enumeration, the contribution to the kinetic energy from expanding the left and right propagators to orders $Q_-, Q_+$, respectively, becomes:

$$\langle H_1(\tau) \rangle^{Q_- Q_+} = \lim_{\beta \to \infty} \sum_{Q=0}^\infty \sum_{\alpha_0 \dots \alpha_Q} \int_0^{\tau_{Q+1} = \beta} d\tau_Q \int_0^{\tau_Q} d\tau_{Q-1} \dots \{ \qquad \} \cdots \int_0^{\tau_2} d\tau_1$$
$$\times \left[ C_{\alpha_\beta}^* C_{\alpha_0} (-1)^{Q-1} e^{-\epsilon_{\alpha_0} \tau_1} \prod_{q=1}^Q e^{-\epsilon_{\alpha_q} (\tau_{q+1} - \tau_q)} H_1^{\alpha_q, \alpha_{q-1}} \right], \quad \text{(D.8)}$$

where the matrix elements and times associated to the states $\alpha', \alpha''$ are now not explicitly written since they'll be contained in the product and summations over kink indices. The empty braces above are used to exaggerate the fact that, upon careful inspection, there is actually one integral missing. That is, there's only $Q - 1$ integrals, rather than $Q$ of them. This missing integral is due to the previously discussed propagators not having the time of the "special kink" $H_1^{\alpha', \alpha''}$ as one of the integration variables. Instead, each of the propagators has integration variables for each of the kinks extending from $\tau = 0$ up to but not including the special kink, and from $\tau = \beta$ down to, but once again not including the special kink. This is also the reason for the exponent of the factor $(-1)^{Q-1}$.

Note also that at the moment, the time $\tau$ has been constant. Instead, we would like to average the kinetic energy over a $\Delta\tau$-sized interval $\tau : [-\Delta\tau/2, \Delta\tau/2]$. This window-averaged

kinetic energy becomes:

$$\langle H_1 \rangle^{Q_- Q_+} = -\frac{1}{\Delta \tau} \lim_{\beta \to \infty} \sum_{Q=0}^{\infty} \sum_{\alpha_0 \ldots \alpha_Q} \int_0^{\tau_{Q+1}=\beta} d\tau_Q \int_0^{\tau_Q} d\tau_{Q-1} \ldots \left\{ \int_{-\Delta\tau/2}^{\Delta\tau/2} d\tau \right\} \ldots$$

$$\times \int_0^{\tau_2} d\tau_1 \left[ C_{\alpha_\beta}^* C_{\alpha_0} (-1)^Q e^{-\epsilon_{\alpha_0} \tau_1} \prod_{q=1}^{Q} e^{-\epsilon_{\alpha_q}(\tau_{q+1}-\tau_q)} H_1^{\alpha_q, \alpha_{q-1}} \right]. \quad \text{(D.9)}$$

We can work with the absolute time of the special kink, as measured from $\tau = 0$, rather than $\tau$, the deviation from $\beta/2$, by performing the substitution $\tau_{q^*} = \beta/2 + \tau$:

$$\langle H_1 \rangle^{Q_+ Q_-} = -\frac{1}{\Delta \tau} \lim_{\beta \to \infty} \sum_{Q=0}^{\infty} \sum_{\alpha_0 \ldots \alpha_Q} \int_0^{\tau_{Q+1}=\beta} d\tau_Q \int_0^{\tau_Q} d\tau_{Q-1} \ldots \left\{ \int_{\Delta\tau/2-\beta/2}^{\Delta\tau/2+\beta/2} d\tau_{q^*} \right\} \ldots$$

$$\times \int_0^{\tau_2} d\tau_1 \left[ C_{\alpha_\beta}^* C_{\alpha_0} (-1)^Q e^{-\epsilon_{\alpha_0} \tau_1} \prod_{q=1}^{Q} e^{-\epsilon_{\alpha_q}(\tau_{q+1}-\tau_q)} H_1^{\alpha_q, \alpha_{q-1}} \right]. \quad \text{(D.10)}$$

By averaging over the time of the special kink $\tau_{q^*}$, there are now a total of $Q$ integrals. Moreover, notice that the kinetic energy starts to look like the configurational weight $W_0(Q, \boldsymbol{\alpha}_Q, \boldsymbol{\tau}_Q)$ from Eq. (27), summed (and integrated) over all worldline configurations. The main difference is the limits of integration of the $\tau_{q^*}$ integral. The integration limits can be rewritten as going from $\tau_{q^*} = 0$ to $\tau_{q^*} = \tau_{q^*+1}$ by multiplying the integrand by the box-car function, which can be defined to be zero for all values of the special kink $\tau_{q^*}$, except inside the interval $\tau_{q^*} : [\Delta\tau/2 - \beta/2, \Delta\tau/2 + \beta/2]$, where it will be one. Formally, the box-car function can be written as:

$$B_{\Delta\tau}(\tau_{q^*}) = H(\tau_{q^*} - \Delta\tau/2 + \beta/2) - H(\tau_{q^*} - \Delta\tau/2 - \beta/2), \quad \text{(D.11)}$$

where $H(x)$ is the Heaviside step-function. By multiplying the integrand of the $\tau_q^*$ integral by this box-car function, the integration limits can be changed the span a larger and arbitrary interval of imaginary times, while leaving the overall result unchanged. Changing the interval to $\tau_q^* \in [0, \tau_{q+1}^*]$:

$$\langle H_1 \rangle^{Q_+ Q_-} = -\frac{1}{\Delta \tau} \lim_{\beta \to \infty} \sum_{Q=0}^{\infty} \sum_{\alpha_0 \ldots \alpha_Q} \int_0^{\tau_{Q+1}=\beta} d\tau_Q \int_0^{\tau_Q} d\tau_{Q-1} \ldots \left\{ \int_0^{\tau_{q^*+1}} d\tau_{q^*} B_{\Delta\tau}(\tau_{q^*}) \right\} \ldots$$

$$\times \int_0^{\tau_2} d\tau_1 \left[ C_{\alpha_\beta}^* C_{\alpha_0} (-1)^Q e^{-\epsilon_{\alpha_0} \tau_1} \prod_{q=1}^{Q} e^{-\epsilon_{\alpha_q}(\tau_{q+1}-\tau_q)} H_1^{\alpha_q, \alpha_{q-1}} \right], \quad \text{(D.12)}$$

where $\tau_{q^*+1}$ denotes the time of the kink after the special kink. Eq. (D.12) is actually only the contribution to the kinetic energy coming from expanding the left propagator to order $Q_-$ and the right one to $Q_+$ in Eq. (D.5). There will be contributions coming from every possible combination of expansion orders of each propagator, with the result looking similar to Eq. (D.12), except with the location of the special kink being different. For example, there are three possible combinations of the propagator expansion orders that lead to $Q = 3$ kinks, and these are $(Q_-, Q_+) : (2, 0), (1, 1), (0, 2)$, for which the special kink (i.e, the one that connects the two propagators) is at $\tau_{q^*} : \tau_1, \tau_2, \tau_3$, respectively. Thus, the total average kinetic energy can be written as:

$$\langle H_1 \rangle = -\frac{1}{\Delta \tau} \lim_{\beta \to \infty} \sum_{Q, \boldsymbol{\alpha}_Q} \int d\tau_Q \left[ \sum_{q^*=1}^{Q} B_{\Delta\tau}(\tau_{q^*}) C_{\alpha_\beta}^* C_{\alpha_0} (-1)^Q e^{-\epsilon_{\alpha_0} \tau_1} \prod_{q=1}^{Q} e^{-\epsilon_{\alpha_q}(\tau_{q+1}-\tau_q)} H_1^{\alpha_q, \alpha_{q-1}} \right]. \quad \text{(D.13)}$$

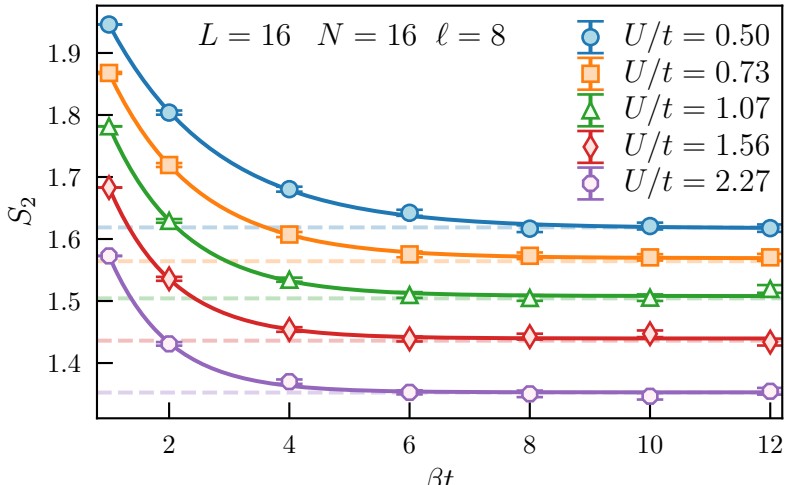

Figure 21: Full second Rényi Entropy for a one-dimensional lattice of $L = 16$ sites at unit-filling under an equal spatial bipartition of size $\ell = 8$. Three-parameter exponential fits in $\beta t$ (solid lines) have been performed on the Monte Carlo results (markers). The dashed horizontal lines denote exact diagonalization values for each of the various interaction strengths. The three-parameter exponential fits the data well and can be used to extrapolate the large $\beta$ limit results of measurements of interest.

Recall that $W_0(Q, \boldsymbol{\alpha}_Q, \boldsymbol{\tau}_Q) = C^*_{\alpha_\beta} C_{\alpha_0} (-1)^Q e^{-\epsilon_{\alpha_0}\tau_1} \prod_{q=1}^{Q} e^{-\epsilon_{\alpha_q}(\tau_{q+1} - \tau_q)} H_1^{\alpha_q, \alpha_{q-1}}$ is the weight of a ground state configuration of worldlines. In terms of these weights, the average kinetic energy is:

$$\langle H_1 \rangle = -\frac{1}{\Delta\tau} \lim_{\beta \to \infty} \sum_{Q, \boldsymbol{\alpha}_Q} \int d\boldsymbol{\tau}_Q \left[ W_0(Q, \boldsymbol{\alpha}_Q, \boldsymbol{\tau}_Q) \sum_{q^*=1}^{Q} B_{\Delta\tau}(\tau_{q^*}) \right] \tag{D.14}$$

$$= -\frac{1}{\Delta\tau} \lim_{\beta \to \infty} \sum_{Q, \boldsymbol{\alpha}_Q} \int d\boldsymbol{\tau}_Q \left[ W_0(Q, \boldsymbol{\alpha}_Q, \boldsymbol{\tau}_Q) N_{\text{kinks}} \right]. \tag{D.15}$$

Recall that the box-car function $B_{\Delta\tau}(\tau_{q^*})$ is one when the special kink is inside the time window $\Delta\tau$, centered around $\beta/2$, and zero otherwise. Thus, the summation $\sum_{q^*=1}^{Q} B_{\Delta\tau}(\tau_{q^*}) \equiv N_{\text{kinks}}$ will measure how many kinks are inside the window for a worldline configuration with $Q$ total kinks. The set of summations and integrals shown above correspond to a sum over all possible ground state configurations of wordlines. The ground state expectaion value of the kinetic energy is therefore:

$$\langle H_1 \rangle_{\text{MC}} \approx -\frac{\langle N_{\text{kinks}} \rangle_{\text{MC}}}{\Delta\tau}, \tag{D.16}$$

where $\langle N_{\text{kinks}} \rangle$ is the average number of kinks inside the window. This expression for the kinetic energy can be easily computed in our path-integral Monte Carlo simulation and is exact in the limit of $\beta \to \infty$.

## E   Large Imaginary Time Projection of Entanglement

Since the time of a simulation scales proportionally with $L^D \beta$, performing measurements with sufficiently small systematic error will become difficult for large systems. To bypass this and

still have a good estimate of the measurement, a three-parameter exponential fit in $\beta$ can be performed, from which the large $\beta$ result can be extrapolated. For the case of the second Rényi Entropy, this fit looks like:

$$S_2(\beta) = S_2^{(\beta \to \infty)} + C_1 e^{-C_2 \beta}, \tag{E.1}$$

where $C_1, C_2$, and $S_2^{(\beta \to \infty)}$ are fitting parameters. The parameter $S_2^{(\beta \to \infty)}$ is the extrapolation of the entanglement entropy in the asymptotic limit of $\beta$. Fig. 21 illustrates the three-parameter exponential fit to $S_2$ data obtained from the PIGS simulation. Various interaction strengths $U/t$ are chosen for this benchmark and the fit works well for all. For each of the interactions, extracting $S_2^{(\beta \to \infty)}$ from the fits gives estimates for the entanglement entropy that are always within one standard deviation of the exact value, computed with exact diagonalization.

# F  Operationally Accessible Entanglement in Terms of $\langle \mathsf{SWAP} \rangle$

The second Rényi accessible entanglement entropy according to Eq. (5) is:

$$S_2^{\mathrm{acc}}(\rho_A) = -2 \ln\left[ \sum_n P_n e^{-\frac{1}{2} S_2(\rho_{A_n})} \right], \tag{F.1}$$

where $\rho_{A_n}$ is the reduced density matrix of spatial partition $A$, projected onto the subspace of local fixed particle number $n$. Recall that one obtains this projected reduced density matrix via Eq. (4): $\rho_{A_n} = \Pi_n \rho_A \Pi_n / P_n$, where $\Pi_n$ are projection operators onto the $n$ subspace and the projected reduced density matrix is normalized by dividing by $P_n$, the probability of measuring a configuration with $n$ particles in subregion $A$. Formally, one can define the projection operators as:

$$\Pi_n = \sum_{a^{(n)}} |a^{(n)}\rangle \langle a^{(n)}|, \tag{F.2}$$

where the summation runs over all possible configurations of the $A$ subregion with fixed local particle number $n$. Recall that the reduced density matrix of subsystem $A$ is obtained by taking the outer product of a wavefunction $|\Psi_0\rangle$ with itself and taking the partial trace with respect to subregion $B$:

$$\rho_A = \sum_b \langle b|\Psi_0\rangle \langle \Psi_0|b\rangle, \tag{F.3}$$

where the summation is carried over the set of possible configurations in $B$.

The ground state of a spatially bipartitioned subsystem can be expressed as a Schmidt decomposition as:

$$|\Psi_0\rangle = \sum_{a,b} C_{ab} |a,b\rangle, \tag{F.4}$$

where the summation is carried over all configurations in $A$ and $B$, and $C_{ab}$ is the complex expansion coefficient for the bipartitioned configuration $|a,b\rangle$. In terms of this Schmidt decomposition, the reduced density matrix becomes:

$$\rho_A = \sum_{a,a'} \left( \sum_b C_{ab} C_{a'b}^* \right) |a\rangle \langle a'|. \tag{F.5}$$

One can now project the reduced density matrix onto the subspace of fixed local particle number $n$ using Eq. (4) and Eq. (F.2):

$$\rho_{A_n} = \frac{1}{P_n} \sum_{a^{(n)}, a'^{(n)}, b^{(N-n)}} C_{a^{(n)} b^{(N-n)}} C_{a'^{(n)} b^{(N-n)}}^* |a^{(n)}\rangle \langle a'^{(n)}|. \tag{F.6}$$

Notice that the sum over $B$ states now runs over configurations that have fixed local particle number $N - n$. This can be done without loss of generality since fixing the local particle number $A$, also fixes the local particle number in $B$. The matrix elements of the projected reduced density matrix can be identified as:

$$\rho_{A_n}\big(a^{(n)}, a'^{(n)}\big) \equiv \big\langle a^{(n)}\big| \rho_{A_n} \big|a'^{(n)}\big\rangle = \frac{1}{P_n} \sum_{b^{(N-n)}} C_{a^{(n)}b^{(N-n)}} C^*_{a'^{(n)}b^{(N-n)}} \, . \tag{F.7}$$

Now that the matrix elements have been identified, the next step is to build the replicated ground state wavefunction, akin to the one that shows up in Eq. (42), but for a ground state projected onto the $n$ local particle number subspace. This can be done by first taking the Schmidt decomposition of a single replica wavefunction shown in Eq. (F.4), expanding it, and regrouping the terms that have same local particle number:

$$|\Psi_0\rangle = \sum_{a^{(0)},b^{(N)}} C_{a^{(0)}b^{(N)}} |a^{(0)}\rangle|b^N\rangle + \sum_{a^{(1)},b^{(N-1)}} C_{a^{(1)}b^{(N-1)}} |a^{(1)}\rangle|b^{N-1}\rangle + \ldots \tag{F.8}$$

Then, acting with the projection operator in Eq. (F.2), only the part of the wavefunction that contributes to the $n$-particle sector is projected out:

$$\big|\Psi_0^{(n)}\big\rangle = \Pi_n |\Psi_0\rangle = \mathcal{N}^{-1} \sum_{a^{(n)},b^{(N-n)}} C_{a^{(n)}b^{(N-n)}} |a^{(n)}\rangle|b^{(N-n)}\rangle \, , \tag{F.9}$$

where $\mathcal{N}^{-1}$ is a normalization constant, as the wavefunction may lose normalization after projection. The replicated ground state in the $n$ subsector is represented by the tensor product of the state above and an identical copy of itself: $\big|\Psi_0^{(n)}\big\rangle \otimes \big|\tilde{\Psi}_0^{(n)}\big\rangle$, where the tilde is used to distinguish between the two replicas. Now acting with the $\text{SWAP}_A$ operator on the replicated ground state, as was done in Eq. (42):

$$\text{SWAP}_A|\Psi_0^{(n)}\otimes\tilde{\Psi}_0^{(n)}\rangle = \sum_{a^{(n)},b^{(N-n)}} \sum_{\tilde{a}^{(n)},\tilde{a}^{(N-n)}} C_{a^{(n)}b^{(N-n)}} D_{\tilde{a}^{(n)}\tilde{b}^{(N-n)}} \big|\tilde{a}^{(n)}\big\rangle \big|b^{(N-n)}\big\rangle \otimes \big|a^{(n)}\big\rangle \big|\tilde{b}^{(N-n)}\big\rangle \, . \tag{F.10}$$

Taking the expectation value in the replicated projected space:

$$\Big\langle \Psi_0^{(n)} \otimes \tilde{\Psi}_0^{(n)} \Big| \text{SWAP}_A \Big| \Psi_0^{(n)} \otimes \tilde{\Psi}_0^{(n)} \Big\rangle =$$
$$\mathcal{N}^{-1} \sum_{a^{(n)},\tilde{a}^{(n)}} \left( \sum_{b^{(N-n)}} C^*_{\tilde{a}^{(n)}b^{(N-n)}} C_{a^{(n)}b^{(N-n)}} \right) \left( \sum_{\tilde{b}^{(N-n)}} D^*_{a^{(n)}\tilde{b}^{(N-n)}} D_{\tilde{a}^{(n)}\tilde{b}^{(N-n)}} \right) , \tag{F.11}$$

where, using Eq. (F.7), the factors in parentheses can be replaced by unnormalized reduced projected density matrix elements:

$$\Big\langle \Psi_0^{(n)} \otimes \tilde{\Psi}_0^{(n)} \Big| \text{SWAP}_A \Big| \Psi_0^{(n)} \otimes \tilde{\Psi}_0^{(n)} \Big\rangle = \mathcal{N}^{-1} \sum_{\tilde{a}^{(n)}} P_n \tilde{P}_n \Big\langle \tilde{a}^{(n)} \Big| \rho_{A_n} \left( \sum_{a^{(n)}} \big|a^{(n)}\big\rangle\big\langle a^{(n)}\big| \right) \rho_{A_n} \Big| \tilde{a}^{(n)} \Big\rangle , \tag{F.12}$$

where the sum over $a^{(n)}$ has been moved near the outer product $\big|a^{(n)}\big\rangle\big\langle a^{(n)}\big|$ to explicitly illustrate that there is now a resolution of the identity operator present. The expectation value of the $\text{SWAP}_A$ operator in the replicated and projected configuration space is thus:

$$\langle \text{SWAP}_{A_n} \rangle \equiv \Big\langle \Psi_0^{(n)} \otimes \tilde{\Psi}_0^{(n)} \Big| \text{SWAP}_A \Big| \Psi_0^{(n)} \otimes \tilde{\Psi}_0^{(n)} \Big\rangle = \mathcal{N}^{-1} \sum_{\tilde{a}^{(n)}} \Big\langle \tilde{a}^{(n)} \Big| \rho_{A_n}^2 \Big| \tilde{a}^{(n)} \Big\rangle , \tag{F.13}$$

where $P_n$ and $\tilde{P}_n$ have been absorbed into the normalization constant. Recognizing that the operation above is a trace, the expectation value can be simplified to

$$\langle \mathrm{SWAP}_{A_n} \rangle = \mathcal{N}^{-1} \operatorname{Tr} \rho_{A_n}^2 \,. \tag{F.14}$$

The second Rényi entanglement entropy for local particle number $n$ becomes:

$$S_2(\rho_{A_n}) = -\ln\left[ \mathcal{N}^{-1} \langle \mathrm{SWAP}_{A_n} \rangle \right] \,. \tag{F.15}$$

Finally, substituting into Eq. (F.1), the $\alpha = 2$ operationally accessible Rényi entanglement entropy becomes:

$$S_2^{\mathrm{acc}}(\rho_A) = -2\ln\left[ \sum_n P_n [\operatorname{Tr} \rho_{A_n}^2]^{1/2} \right] = -2\ln\left[ \sum_n P_n \left[ \mathcal{N} \langle \mathrm{SWAP}_{A_n} \rangle \right]^{1/2} \right] \,. \tag{F.16}$$

This expression is now amenable to estimation in the `PIGSFLI` algorithm.

# G  Improved accessible entanglement estimator given limitations of sampling particle number sectors

In practice, the probability of measuring most local particle number sectors could be small, and many of these sectors may not be visited in a given simulation. This can cause the argument of the natural logarithm in Eq. (51) to become vanishingly small, and the accessible entanglement estimator to be undefined. Additionally, if $P_n$ is finite but very small, configurations with a certain number of SWAP kinks in that $n$-sector will not be sampled, again making Eq. (51) undefined.

To tackle this situation in practice, the summation in Eq. (51) can be expanded into two terms: the sum of all particle number sectors $n$ that were measured more frequently in the simulation and have good statistics, and the contribution from $\tilde{n}$-sectors that were only rarely measured with poor statistics:

$$S_2^{\mathrm{acc}}(\rho_A) \approx -2\ln\left[ \sum_n P_n \left[ \operatorname{Tr} \rho_{A_n}^2 \right]^{1/2} + \sum_{\tilde{n}} P_{\tilde{n}} \left[ \operatorname{Tr} \rho_{A_{\tilde{n}}}^2 \right]^{1/2} \right] \,. \tag{G.1}$$

In practice, the second term can be discarded without consequence since it is negligible compared to the statistical error of the first term. Using the expansion

$$\ln(x + K) \approx \ln(x) + \frac{x}{K} - \mathcal{O}(x^2) \,, \tag{G.2}$$

the accessible entanglement in Eq. (G.1) can be approximated as:

$$S_2^{\mathrm{acc}}(\rho_A) \approx -2\left[ \ln \sum_n P_n \left[ \operatorname{Tr} \rho_{A_n}^2 \right]^{1/2} + \frac{\sum_{\tilde{n}} P_{\tilde{n}} \left[ \operatorname{Tr} \rho_{A_{\tilde{n}}}^2 \right]^{1/2}}{\sum_n P_n \left[ \operatorname{Tr} \rho_{A_n}^2 \right]^{1/2}} \right] \,. \tag{G.3}$$

Since $\operatorname{Tr} \rho_{A_n}^2 \leq 1$ and $P_{\tilde{n}} < 1$, the second term is bounded from above by the sum of "bad" (i.e., poorly measured) probabilities:

$$\sum_{\tilde{n}} P_{\tilde{n}} \left[ \operatorname{Tr} \rho_{A_{\tilde{n}}}^2 \right]^{1/2} \Big/ \sum_n P_n \left[ \operatorname{Tr} \rho_{A_n}^2 \right]^{1/2} < \sum_{\tilde{n}} P_{\tilde{n}} \,. \tag{G.4}$$

The first term of Eq. (G.3) is computed as a QMC average, with some associated statistical error $\sigma$. Thus the quantity $\sum_{\tilde{n}} P_{\tilde{n}}$ can be compared to this statistical error and discarded if it is much smaller in comparison. To confirm this, a ratio between the so-called "throwaway error" and statistical error bar may be computed to confirm the relative accuracy of the procedure. For example, for the results presented in this work, the contribution to the second accessible Rényi Entropy coming from the poorly sampled $n$-sectors was thrown out if: $\sum_{\tilde{n}} P_{\tilde{n}} / \sigma < 0.1$.

In summary, the above scheme allows for the calculation of the second accessible Rényi entanglement entropy by discarding data coming from poorly sampled $n$-sectors that can result in undefined results.

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
