# Peer review of "A Path Integral Ground State Monte Carlo Algorithm for Entanglement of Lattice Bosons"

_SciPost Physics, doi:SciPost Phys. 14, 054 (2023)_

## Round 2 · Referee Report · Anonymous (Referee 1) · 2022-9-7

Strengths

  1. A very detailed description of the algorithm and of the Monte Carlo estimators
  2. An implementation of important quantities that are not contained in other open source packages. It can also compute the symmetry-resolved and the accessible entanglement entropy.
  3. an open source implementation

Weaknesses

  1. new physics content is lacking
  2. A very large part of the paper (the first 15 pages plus large parts of the appendix) is not new.

Report

See below for the requested changes.
The argument in favor of publication in Physics is that the Monte Carlo estimators plus an implementation of the symmetry-resolved and the accessible entanglement entropies are provided, which have not been published before as far as I know.
The argument for publication in, eg, CodeBases is that there is little new physics content.

Requested changes

  1. Regarding expressions (41) and similar ones throughout the text: If we are not in the Mott insulating regime, the system is gapless in the thermodynamic limit. Hence, the exponential scaling is a finite size effect for the two lowest values of U that are considered. For values of L \gg 8, I would expect that this exponential law is almost impossible to see numerically. This should be clarified in the text.

  2. It would be useful to have an artistic 3d-like figure depicting the two replicas in imaginary time for a 1d system.

  3. Throughout the text, the authors write U/t but never \beta t. This should be made consistent

  4. In fig 10, the inset shows that the data are compatible with a log behavior. Can any further quantitative information be extracted from it, given the error bars?

  5. (i) For the 1d quantum phase transition, the authors should check the literature more carefully for the location of the quantum critical point. (ii) How should the entanglement entropy behave for such a Kosterlitz-Thouless transition? Can a finite size analysis based on the Renyi entropy be applied to determine the quantum critical point more accurately?

  6. I would like to know how the error bars are determined, on (i) <SWAP> and (ii) the logarithm of the latter quantity (which is the non-linear operation necessary to compute the entanglement entropy) -- and what is the distribution of these measurements?

  7. The output in Appendix A is not comprehensible to me. The readme.md file of the git repo shows the same output, and I did not get much wiser from it. The documentation of the code in the git repo should be improved substantially. Is the code working on 64 bit architectures?

  8. I did my best to find a typo: there is a verb missing in the 4th line of the caption of fig 4. The authors deserve praise for writing so precisely.

  • validity: top
  • significance: high
  • originality: good
  • clarity: top
  • formatting: perfect
  • grammar: perfect

Author:  Adrian Del Maestro  on 2022-10-21  [id 2946]

(in reply to Report 1 on 2022-09-07)
Category:
remark
answer to question
objection
reply to objection
correction

We thank the referee for their careful review and thoughtful comments.

Based on the comments of the referee, we acknowledge that the resulting rather long paper, combined with its overly technical title and abstract, may have "buried the lead." Motivated by previous and on-going experimental searches able to measure entanglement entropy in ultracold lattice bosons, we aimed to devise an exact ground state numerical method able to make direct predictions of quantum information quantities in these systems. Numerical methods previously reported in the literature for itinerant lattice soft-core bosons to date are based on exact diagonalization or the density matrix renormalization group (DMRG) and are thus limited to small systems or one spatial dimension (or quasi-1D, highly anisotropic 2D systems). DMRG studies also require imposing a local restriction on the bosonic Hilbert space that can lead to large errors in entanglement for weak interactions. Previous quantum Monte Carlo methods for entanglement work only for distinguishable local degrees of freedom (e.g. spin systems) or within a continuous space path integral framework. Moreover, the lack of any open source quantum Monte Carlo code applicable to extended Bose-Hubbard systems has prevented ab initio studies of experimentally accessible quantum simulators. Our intent was to write a pedagogical manuscript on the general algorithim and method, while at the same time, highlighting our novel results on both spatial mode and symmetry resolved entanglement in the Bose-Hubbard model. The latter has generated considerable interest in recent years due to its connection to the accessible entanglement in a quantum many-body phase that could be utilized as a resource for quantum information processing applications. The resulting length of our submitted manuscript (>50 pages), combined with an admitted lack of emphasis on the new results, led the referee to question the suitability of our paper for SciPost Physics.

We disagree with this sentiment. The new contributions of this work include:

  1. A ground state QMC algorithm that, while potentially understood by experts, had never appeared in the literature. We derived new Monte Carlo estimators able to measure Rényi entanglement entropies within the lattice path integral framework.
  2. New results for the spatial entanglement entropy in the 1D and 2D Bose-Hubbard model both at the quantum critical point, and across the superfluid-insulator phase diagram for considerably larger system sizes than had been previously measured (up to 1024 bosons in 2D).
  3. An analysis of subleading corrections to the area law at the 2D quantum critical point. In the current version of the manuscript, we now substantially improve our analysis here (motivated by the referee) and are able to extract the number of Goldstone modes in the resulting gapless phase.
  4. Particle number distributions and the symmetry resolved entanglement in the superfluid, critical, and Mott insulating regimes for a small system size (N = 8 particles, Figure 13). We interpreted the observed strong n dependence in the Mott phase in terms of the interaction between holes and doublons. To further explore this physics, we have performed additional simulations for a much larger system (N = 64) far outside the reach of exact diagonalization, and included a new Figure 14.

We now better highlight the physical focus and new results in the substantially modified introduction.

Focusing on the SciPost Physics Acceptance Criteria, we believe our paper "Open(s) a new pathway in an existing or a new research direction, with clear potential for multipronged follow-up work."

While the SWAP estimator had been previously implemented in other quantum Monte Carlo flavors, this has only included those with localized and distinguishable degrees of freedom, e.g. spin models or in the spatial continuum. The extension to lattice models of itinerant bosons that are clearly the most relevant for experimental studies of entanglement (see e.g. Refs. [17-18]) had yet to be accomplished. As evidenced by the lengthy derivation of the SWAP in terms of weights in the continuous imaginary time quantum Monte Carlo algorithm presented here, the generalization to this case is non-trivial. Unlike a spin system, the number of degrees of freedom (worldlines) in a region defined by some subset of spatial modes is a dynamical quantity. As mentioned above, while entanglement in the Bose-Hubbard model has been previously studied in DMRG, these simulations were limited to 1D systems and always implemented a local occupation number restriction (sometimes of only a few bosons per site), resulting in potentially extremely larger errors (of order 100%) in the weakly interacting regime (see e.g. https://journals.aps.org/prb/abstract/10.1103/PhysRevB.105.L121116).

Another benefit of the ground state QMC method presented here is its ability to scale to systems in general dimensions, and based on the comment of referee 1, we have more closely analyzed the sub-leading logarithmic correction to the area law in the 2D Bose-Hubbard model in a 32x32 system composed of 1024 bosons at unit filling. It is predicted that logarithmic corrections should arise due to the existence of a spontaneously broken continuous symmetry in the thermodynamic limit, contributing a universal coefficient due to the presence of a low-energy "tower of states" spectrum and a Goldstone boson. The universal prefactor of this term is known to be related to the number of Goldstone modes, and we report that the value we extract from the QMC is consistent with the value of one for the Bose-Hubbard model. This represents an important first step in studying these, and higher order corrections to the area law which can encode information on the central charges that characterize higher-dimensional conformal field theory, believed to be the fundamental constants that quantify how entropy monotonically decreases under renormalization group flows.

In addition, the growing recent interest in the symmetry resolved entanglement (the number of papers with this in the title/abstract has doubled every year for the last few years) is a strongly motivating factor for our work. The vast majority of these studies have been limited to non-interacting models, as there was simply no technology or method available beyond exact diagonalization for its measurement in interacting systems until now. Computing this quantity, which can have a very large signal to noise ratio in quantum Monte Carlo is no small task, and utilizing the presented algorithm we believe that there is clear potential for follow up work. This may be especially important given the tendency of this quantity to signal quantum phase transitions and its experimental measurement through post-selection. It can also display highly non-trivial dependence on the number of particles in the subsystem (n) and we accentuate this with a new figure 14 in a system of N = 64 bosons.

Response to specific report and requested changes:

We would like to thank the referee for their insightful comments. We will address them here and have edited the paper accordingly, resulting in an improved manuscript.

Referee Writes:

(1) Regarding expressions (41) and similar ones throughout the text: If we are not in the Mott insulating regime, the system is gapless in the thermodynamic limit. Hence, the exponential scaling is a finite size effect for the two lowest values of U that are considered. For values of L \gg 8, I would expect that this exponential law is almost impossible to see numerically. This should be clarified in the text.

Our Response: We do not agree with the referee about the absence of exponential scaling of estimators in the superfluid (gapless) phase. In a finite size system, the gap will indeed decrease as a polynomial in the inverse system size. Once β is larger than the inverse gap, the energy will decay approximately as a single exponential in the projection or imaginary time length β. This is true both for T=0 and T>0 algorithms. As an example, we include in this reply a plot of the second Rényi Entanglement Entropy, S_2, obtained with our algorithm at various values of the interaction strength U/t in the superfluid phase for a larger system size of N=256 bosons under an equal bipartition of ℓ=128 sites in one dimension. We use the three-parameter fitting shown in Eq. (54) of the paper. To demonstrate this, we have linearized by plotting the results as a function of the exponential dependence: e^{-C_2 β t}.

See: supplemental_figures.pdf (1)

Related changes: When introducing the fits of Eqs. (41) and (54), which depend exponentially on the projection length, we now clarify the origin of the exponential behavior.

Referee Writes:

(2) It would be useful to have an artistic 3d-like figure depicting the two replicas in imaginary time for a 1d system.

Our Response: We thank the referee for the suggestion. Having a diagram of this very non-trivial space should help beginners and experts alike.

Related changes: Added 3d diagram illustrating the two-replica configuration space in which we measure second Rényi entropies.

Referee Writes:

(3) Throughout the text, the authors write U/t but never \beta t. This should be made consistent

Our Response: We agree. In practice, we set the tunneling parameter t = 1.0 in the code.

Related changes: For consistency, we now report a rescaled projection length β → βt when reporting results in terms of the dimensionless interaction strength U/t.

Referee Writes:

(4) In fig 10, the inset shows that the data are compatible with a log behavior. Can any further quantitative information be extracted from it, given the error bars?

Our Response: We thank the referee for bringing up this point, which we were originally planning to save for a future publication on the issue. A very interesting piece of quantitative information that can be extracted from this inset would be the universal logarithmic correction to the Rényi entropy. This correction is known to be directly proportional to the number of Goldstone modes N_G where: S_2 = aℓ + (N_G/2) ln(ℓ) + …

From the fit reported in the manuscript for square subregions with linear size 5 up to 20, we find a logarithmic correction of b = 0.44 +/- 0.08 = N_G/2. This is consistent with the number of Goldstone modes in the Bose-Hubbard model, N_G = 1.

For the largest system sizes we studied, statistical uncertainties are still large and further work needs to be done to better quantify the nature of this sub-leading logarithmic correction to the area law. This represents an exciting potential future application of our algorithm which we plan to pursue.

Related changes: Numerical values of the fit parameters a,b,c are now reported. The universality of the logarithmic correction is now mentioned and its direct proportionality to the number of Goldstone modes. Relevant literature has been cited.

Referee Writes:

(5) (i) For the 1d quantum phase transition, the authors should check the literature more carefully for the location of the quantum critical point. (ii) How should the entanglement entropy behave for such a Kosterlitz-Thouless transition? Can a finite size analysis based on the Rényi entropy be applied to determine the quantum critical point more accurately?

Our Response:

(i) Determining the exact location of this critical point is, not surprisingly, a hard task, with many values near (U/t)_c \approx 3.3 being reported in the literature that have been obtained via different methods (see e.g. Fig. 3 in the supplemental material of https://journals.aps.org/prl/abstract/10.1103/PhysRevLett.108.116401). For finite systems such as the ones we report on, a value of (U/t)_c \approx 3.3 is close enough to the quantum phase transition that critical behavior can be observed.

ii) We expect that a finite-size analysis on the Rényi entropies should help compute the quantum critical point more accurately. A similar approach was taken in Figure 2 of https://doi.org/10.1088/1742-5468/2008/05/P05018 using a measure based on the von Neumann entropy. Moreover, we also expect the accessible entanglement entropies to be of particular help in this endeavor, as previous exact diagonalization and DMRG studies (https://journals.aps.org/pra/abstract/10.1103/PhysRevA.93.042336, https://journals.aps.org/pra/abstract/10.1103/PhysRevA.100.022324, and https://doi.org/10.1103/PhysRevB.105.L121116) show that near quantum phase transitions this quantity has a peak that appears to scale. One of our goals with the development of this algorithm is to ultimately perform such finite-size analyses on the various entanglement measures for quantum critical point estimation.

Related changes: We have added citations to large number of references of these one-dimensional critical point estimations. We also now comment on how a finite size scaling analysis of our S_2 vs. U/t results for large systems could be used to identify the critical point.

Referee Writes:

(6) I would like to know how the error bars are determined, on (i) <SWAP> and (ii) the logarithm of the latter quantity (which is the non-linear operation necessary to compute the entanglement entropy) -- and what is the distribution of these measurements?

Our Response: To avoid dealing with error propagation through the non-linear functions involved, we compute the error bars of the second Rényi entropies via the jackknife resampling method.

The <SWAP> estimator can be obtained by counting the number of times that m SWAP kinks are measured in a subregion of linear size ℓ, divided over the number of times that zero SWAP kinks were measured. We first compute the average value of this estimator over the entire data set. We then obtain the i-th jacknife estimate by subtracting out the i-th data point from the average of the entire data set. Each of the i-th data points where obtained from independent simulations with different random seeds. One can also then define a jackknife estimate for a function. This is done by taking the jackknifed dataset computed above and applying the desired function to each element. In our case, we take the negative log of each element of the jacknifed data set above.

Finally, the average value of the Rényi entropy is computed by averaging the jacknifed S_2 data set. The jackknife error bars are then obtained as the standard deviation of the jacknifed S_2 data set multiplied by the square root of the number of samples.

A figure showing the distribution of the jacknifed S2 samples for various interaction strengths is included in this reply.

See: supplemental_figures.pdf (2)

Related changes: We now mention that error bars are computed using the jackknife method in the main text.

Referee Writes:

(7) The output in Appendix A is not comprehensible to me. The readme.md file of the git repo shows the same output, and I did not get much wiser from it. The documentation of the code in the git repo should be improved substantially. Is the code working on 64 bit architectures?

Our Response: In Stage (1/3) of the simulation, the goal is to identify a chemical potential (here just an algorithmic parameter in the canonical ensemble) which yields an average number of particles close the desired one. The output shows histograms of the total particle number distribution. Each of the asterisks (*) represents a normalized count. For canonical ensemble simulations, like the one shown above, the only particle numbers visited are N-1, N, and N+1, where N is the target number of particles fixed via command line options. A grand canonical simulation will show histograms with more particle numbers than these. Once the peak of the distribution is at N, and it's at least 33% larger than the next largest sector, we proceed to the "fine tuning η" stage. Here, η, the worm fugacity, is iteratively adjusted until we reach the desired percentage of configurations with no worms present (currently, we set this window between 40 and 45%).

Stage (2/3) is burn-in, with the Markov chain being equilibrated, with the number of sweeps set from the command line.

Finally, in Stage (3/3) measurements are performed and samples collected. Once the desired number of samples are collected, the simulation stops. Near the bottom of the terminal output above, the number of times that each of the updates is accepted and proposed are shown as a fraction. The number of times that the update is accepted is shown in the numerator, whereas the times that it was proposed is shown in the denominator.

The total run time of equilibration and Main Monte Carlo loops is shown at the bottom of the output, in seconds.

The code is working on 64 bit architectures.

We have added the output explanation above both to the code documentation and also to the appendix A of the paper.

Related changes: The terminal output is now thoroughly explained in Appendix A and the code's documentation.

Referee Writes:

(8) I did my best to find a typo: there is a verb missing in the 4th line of the caption of fig 4. The authors deserve praise for writing so precisely.

Our Response: We were lucky to have various members of the Del Maestro Research group help us proofread the manuscript. We thank the referee on praising the precision on writing.

Related Changes: The mentioned typo has been fixed.

Attachment:

supplemental_figures_ckjnySP.pdf

---

## Round 2 · Referee Report · Anonymous (Referee 2) · 2022-9-21

Strengths

  1. Well-written paper, very comprehensive and self-contained introduction to WA, Swap operation to compute EE etc
  2. Accompanying code is a clear added value
  3. The extension to T=0 allows to improve statistics for the EE

Weaknesses

  1. There is no new physics
  2. The numerical techniques highlighted (path integral using the worm algorithm WA, computations of entanglement entropy EE using the SWAP operator) are well known, and the only novelties (modifications to use the WA at T=0, computation of the accessible and symmetry-resolved EE) are too minor to justify publication in SciPost Phys

Report

Everything is explained in the weak/strong points above: the paper as it stands does not contain any novelty that could justify publication in SciPost in my opinion (see below) but it is certainly very useful as a companion to the released Quantum Monte Carlo code.

I think SciPost Phys. is not the appropriate review for this paper, but I think this manuscript+code would be an excellent fit to SciPost Physics CodeBase.

On the lack of novelty : - The discussion of how to modify non-local updates (such as performed by the WA) to be able to perform computations directly at T=0 can be found in several papers (e.g. https://doi.org/10.1007/978-3-642-35106-8_7, https://doi.org/10.1103/PhysRevB.82.024407 and perhaps others in other different forms) To the best of my knowledge, it has never been exactly presented for the worm algorithm, and it is good that the authors do that, but it is not a major change that justifies publication in itself (QMC practitioners certainly know / knew how to do this). - The same holds for the extra Swap updates presented in Sec. 5 to measure entanglement entropy within the worm algorithm, or slight modifications to measure the accessible and symmetry-resolved EE

Additional remarks: - Let me point out one related manuscript (https://doi.org/fr/10.1103/PhysRevLett.86.5164) that shows how to perform calculations directly in the thermodynamic limit and/or T=0 using the loop algorithm, using improved estimators. I think it is relevant for the discussion as improved estimators have been proposed for entanglement entropy (see e.g. Eq. 48 or https://doi.org/10.1103/PhysRevB.92.115146) and other interesting quantities in QMC. - I did not check all length derivations in the appendix, but assume they are correct as the final results (e.g. on energy estimator) is correct - In this very long (58 pages!) paper, I only spotted one typo (p5 : the sentence starting by ``The accessible entanglement entropy, which bounds…’’ is not finished), but I may have missed others.

Requested changes

  1. Submit to SciPost Phys Codebase

  • validity: top
  • significance: ok
  • originality: low
  • clarity: top
  • formatting: perfect
  • grammar: perfect

Author:  Adrian Del Maestro  on 2022-10-21  [id 2944]

(in reply to Report 2 on 2022-09-21)

We thank the referee for reading the manuscript and providing their comments. We have provided a general response to concerns about physics results in the report to Referee 1 and we acknowledge that we did not do an appropriate job of highlighting these results in our lengthy manuscript.

Here we respond to their specific comments.

Referee Writes:

The discussion of how to modify non-local updates (such as performed by the WA) to be able to perform computations directly at T=0 can be found in several papers (e.g. https://doi.org/10.1007/978-3-642-35106-8_7, https://doi.org/10.1103/PhysRevB.82.024407 and perhaps others in other different forms) To the best of my knowledge, it has never been exactly presented for the worm algorithm, and it is good that the authors do that, but it is not a major change that justifies publication in itself (QMC practitioners certainly know / knew how to do this).

Our Response: The references the referee points to are for Stochastic Series Expansion QMC which is only applicable to lattice models with localized degrees of freedom (e.g. spin models), not to models of itinerant bosons where the local Hilbert space is extensive. We are not arguing that the presentation of a T=0 Bose-Hubbard QMC code alone is sufficient justification for publication in SciPost Physics alone; rather, we believe the the entirety of the manuscript, including the presentation of a method for computing symmetry resolved entanglement entropies in Bose-Hubbard systems (without a local particle number constraint) justifies publication in SciPost Physics.

Referee Writes:

The same holds for the extra Swap updates presented in Sec. 5 to measure entanglement entropy within the worm algorithm, or slight modifications to measure the accessible and symmetry-resolved EE"

Our Response: We disagree with the referee that the presentation of the SWAP estimator in the context of the Bose-Hubbard model does not justify publication. Presently the largest scale experimental efforts to measure entanglement entropy in interacting many-body systems is in ultracold atom systems which are well modeled by the Bose-Hubbard model. To date, numerical studies of entanglement entropy in the Bose-Hubbard model have been limited to applying DMRG to quasi-1D geometries; this approach requires artificially constraining the local Hilbert space which can introduce substantial systemic errors to the symmetry resolved entanglement entropy (see e.g. https://journals.aps.org/prb/abstract/10.1103/PhysRevB.105.L121116).

Therefore, the presentation of this algorithm in the literature is relevant to ongoing and future experiments probing entanglement in many-body systems. We view lack of prior presentations of either this algorithm and or resulting studies of Bose-Hubbard systems to be evidence that this experimentally relevant method is a non-trivial contribution to the literature.

Response to additional remarks made by the referee:

Referee Writes:

Let me point out one related manuscript (https://doi.org/fr/10.1103/PhysRevLett.86.5164) that shows how to perform calculations directly in the thermodynamic limit and/or T=0 using the loop algorithm, using improved estimators. I think it is relevant for the discussion as improved estimators have been proposed for entanglement entropy (see e.g. Eq. 48 or https://doi.org/10.1103/PhysRevB.92.115146) and other interesting quantities in QMC.

Our Response: We appreciate the referee's suggestion.

Referee Writes:

I did not check all length derivations in the appendix, but assume they are correct as the final results (e.g. on energy estimator) is correct

Our Response: We appreciate the referee's confirmation that the final results are correct.

Referee Writes:

In this very long (58 pages!) paper, I only spotted one typo (p5 : the sentence starting by The accessible entanglement entropy, which bounds…’’ is not finished), but I may have missed others.

Our Response: We appreciate the referee pointed out this typo. It has been corrected in the resubmitted manuscript.

---

## Round 2 · Referee Report · Anonymous (Referee 3) · 2022-9-26

Strengths

The paper gives a thorough overview of the literature and the relevant references on Path Integral Ground State Monte Carlo and the recent Developments.
Also of clear value is the inclusion of the measurement of entanglement entropies within the algorithm and the explanation of measurements at T=0.
All Formula that are required for the code are thoroughly derived and therefore
the paper provides a good self-contained overview of the subject.

Weaknesses

I concur with the other referees that no new physics is contained in this article
and the literature already given by the other referee, shows that the GS extension and the entanglement entropies, have already been discussed elsewhere.

Report

The lack of novelty does not make this suitable for scipost physics, but this article is a suitable supplement to PIGSFLI. Therefore I recommend to submit the PIGSFLI project to scipost codebase with the improved companion article. If the project is submitted to SciPost Codebase. In that case, there's some more work to do on the codebase:

  • The author's claim, that the code is open source, but neither in the paper, nor in the github repo, I could find a hint of which open source license they used.

  • portability seems to not have been a top concern. Currently, PIGSFLI does not compile on g++-12 (trivial change), clang++ in contrast reports numerous suspicious errors:

result of '2^32' is 34; did you mean '1LL << 32'? [-Wxor-used-as-pow] return randInt((2^32)-1);

The issue here is that "^" is the XOR operator in C/C++ and not exponentiation as in other languages. Without proper documentation of randint it is hard to tell for sure whether this is intended, but assuming that randint(int n) produces integers from [0..n] this function would only produce ints from [0..34] and not over the entire range of a 32bit int as probably intended... Also the hardcoding of 32bits here can certainly be improved.

It might be worthwhile to look into github actions, in order to set up pipelines that catch these errors early.(https://github.com/features/actions)

Which brings us to documentation. Currently the codebase lacks a lot of in-code documentation, and therefore I recommend to use some standard, e.g. doxygen (https://doxygen.nl/manual/docblocks.html)

  • In addition to the inclusion of a LICENSE.md file, I recommend the inclusion of a Citation file format file (https://citation-file-format.github.io/) and in the open source spirit, a contribution guide (CONTRIBUTING.md)

Requested changes

I here give a list of typos/errors that I found in the paper for the authors to consider:
2.1 : The sentence "The accessible entanglement entropy, which bounds the operationally accessible entanglement in the presence of such a superselection rule." misses sth. after the relative clause
3.2 : "average number of particle" -> "average number of particles"
3.5: "to visit all worldline configuration" -> "to visit all worldline configurations"
"proposing 2" -> "proposing two"
4.2 "ground state consisting for both" -> "ground state consisting of both"
4.3 "entropes" -> "entropies"
5.1.1 "etc \newline ..." -> make sure that "etc..." is not broken, e.g. "etc~\dots"
5.1.1 Item 1. in enumeration misses a termination point.

Appendix A: "We have chosen the course tuning" -> "We have chosen the coarse tuning".

In the entire appendix, termination points behind equations are placed inconsistently. Eq. 89 has one, whereas Eq. 91 lacks it.

In the references section, "Rényi" and "Bose-Hubbard" are written inconsistently .
And Ref. 116 probably should have "lieb-liniger" capitalized.

Caption Fig. 12: "correspond to interactions strengths" -> "correspond to interaction strengths"
Caption Fig. 14: "insidet" -> "inside"

  • validity: top
  • significance: good
  • originality: low
  • clarity: good
  • formatting: excellent
  • grammar: perfect

Author:  Adrian Del Maestro  on 2022-10-21  [id 2945]

(in reply to Report 3 on 2022-09-26)
Category:
remark
answer to question
reply to objection

We would like to thank the referee for their very extensive list of typos found and the suggestions related to improving aspects of our code as an open source project. Below we address some of the comments from the "Report" and "Requested Changes" section.

Referee Writes:

The author's claim, that the code is open source, but neither in the paper, nor in the github repo, I could find a hint of which open source license they used.

Our Response: We have now included an MIT open source license to the code repositories.

Referee Writes:

portability seems to not have been a top concern. Currently, PIGSFLI does not compile on g++-12 (trivial change)

Our Response: We thank the referee for pointing this out as one of our ambitions with this still infant code is for it to have good portability. We had tested on an older version of g++ and it had compiled successfully. After the referee's observation, we realized that for g++-12 it indeed was not compiling but we were able to easily fix this by adding an "#include <memory>" statement to the RNG.h header file. The code now compiles with g++-12.

Referee Writes:

...clang++ in contrast reports numerous suspicious errors: result of '2^32' is 34; did you mean '1LL << 32'? [-Wxor-used-as-pow] return randInt((2^32)-1);

Our Response: We would like to clarify that the output above was from a warning and not an error. This warning was coming from the randInt() function defined in the library that we are using for random number generation (see the RNG.h header file). Since we are not using this function, it did not affect code results. These "return randInt((2^32)-1)" statements were originally written as placeholder values by the developers of RNG.h but have now been fixed such that the aforementioned function returns a random integer within the full range of the RNG engine, as the referee had suspected was intended.

Additionally, we have fixed all other issues that were causing compiler warnings. After testing with clang++ and g++-12, no warnings should be given now.

Referee Writes:

I here give a list of typos/errors that I found in the paper for the authors to consider: 2.1 : The sentence "The accessible entanglement entropy, which bounds the operationally accessible entanglement in the presence of such a superselection rule." misses sth. after the relative clause 3.2 : "average number of particle" -> "average number of particles" 3.5: "to visit all worldline configuration" -> "to visit all worldline configurations" "proposing 2" -> "proposing two" 4.2 "ground state consisting for both" -> "ground state consisting of both" 4.3 "entropes" -> "entropies" 5.1.1 "etc \newline ..." -> make sure that "etc..." is not broken, e.g. "etc~\dots" 5.1.1 Item 1. in enumeration misses a termination point. Appendix A: "We have chosen the course tuning" -> "We have chosen the coarse tuning". In the entire appendix, termination points behind equations are placed inconsistently. Eq. 89 has one, whereas Eq. 91 lacks it. In the references section, "Rényi" and "Bose-Hubbard" are written inconsistently . And Ref. 116 probably should have "lieb-liniger" capitalized. Caption Fig. 12: "correspond to interactions strengths" -> "correspond to interaction strengths" Caption Fig. 14: "insidet" -> "inside"

Our Response: We thank the referee for finding all of these typos. They have all been fixed.

Finally, we appreciate the recommendation of tools such as doxygen and GitHub actions. As we keep developing our code for even more applications, these tools will be considered to facilitate development and make the experience for new users as accessible as possible.

---

## Round 3 · Referee Report · Anonymous (Referee 3) · 2022-11-4

Strengths

  • High level of detail
  • Reproducible using the associated Open Source package

Weaknesses

  • Length

Report

I still highly praise the author's for their detailed exposition of the algorithm(I think more algorithms out there should have similar descriptions in the public literature). As the author's have pointed out in their reply, the presence of a Code enabling the measurement of ground-state symmetry resolved entanglement in the presence of interactions presents a step beyond ED and DMRG possibilities and hence meets one of SciPost Physics Acceptance criteria("Open a new pathway in an existing or a new research direction").

Requested changes

One more typo: - Page 25: "To stress that larger system sizes are needed ... , the result[s] have been plotted..."

---

## Round 3 · Referee Report · Anonymous (Referee 1) · 2022-11-5

Strengths

  • great detail
  • open source repository

Weaknesses

the high level of detail comes at the expense of a very long, perhaps too long, paper

Report

I am satisfied with the changes implemented by the Authors. I think the physics content has improved (cf the discussion of the number of goldstone modes at the quantum critical point of the 2D Bose-Hubbard model), and I thank the authors for the additional "artistic" figure which looks very good. The coding issues mentioned by another referee have also improved.
This work provides a very valuable contribution to quantum Monte Carlo simulations and the computation of entanglement entropies in particular.

---

## Round 3 · Referee Report · Anonymous (Referee 2) · 2022-11-14

Strengths

  • Details of algorithms are very well explained
  • Accompanying code
  • Excellent pedagogical style of writing

Weaknesses

  • Lack of new physics

Report

I have read in detail the answers to the referees and the new manuscript.
I appreciated the effort of the authors to reply to all points in detail.
I continue to have a different point of view with the authors on the existence of really new aspects (methodology, physics) presented in this work, and to think that SciPost Physics CodeBase would be the proper arena for this manuscript+code.
However I do not want to block the authors of an otherwise scientifically good and excellently written paper -- so I can also suggest to publish the current version of the manuscript in SciPost Physics.

---

## Round 3 · Author Response

Based on the comments of the referee, we acknowledge that the resulting rather long paper, combined with its overly technical title and abstract, may have "buried the lead." Motivated by previous and on-going experimental searches able to measure entanglement entropy in ultracold lattice bosons, we aimed to devise an exact ground state numerical method able to make direct predictions of quantum information quantities in these systems. Numerical methods previously reported in the literature for itinerant lattice soft-core bosons to date are based on exact diagonalization or the density matrix renormalization group (DMRG) and are thus limited to small systems or one spatial dimension (or quasi-1D, highly anisotropic 2D systems). DMRG studies also require imposing a local restriction on the bosonic Hilbert space that can lead to large errors in entanglement for weak interactions. Previous quantum Monte Carlo methods for entanglement work only for distinguishable local degrees of freedom (e.g. spin systems) or within a continuous space path integral framework. Moreover, the lack of any open source quantum Monte Carlo code applicable to extended Bose-Hubbard systems has prevented ab initio studies of experimentally accessible quantum simulators. Our intent was to write a pedagogical manuscript on the general algorithim and method, while at the same time, highlighting our novel results on both spatial mode and symmetry resolved entanglement in the Bose-Hubbard model. The latter has generated considerable interest in recent years due to its connection to the accessible entanglement in a quantum many-body phase that could be utilized as a resource for quantum information processing applications. The resulting length of our submitted manuscript (>50 pages), combined with an admitted lack of emphasis on the new results, led the referee to question the suitability of our paper for SciPost Physics.

We disagree with this sentiment. The new contributions of this work include:

  1. A ground state QMC algorithm that, while potentially understood by experts, had never appeared in the literature. We derived new Monte Carlo estimators able to measure Rényi entanglement entropies within the lattice path integral framework.
  2. New results for the spatial entanglement entropy in the 1D and 2D Bose-Hubbard model both at the quantum critical point, and across the superfluid-insulator phase diagram for considerably larger system sizes than had been previously measured (up to 1024 bosons in 2D).
  3. An analysis of subleading corrections to the area law at the 2D quantum critical point. In the current version of the manuscript, we now substantially improve our analysis here (motivated by the referee) and are able to extract the number of Goldstone modes in the resulting gapless phase.
  4. Particle number distributions and the symmetry resolved entanglement in the superfluid, critical, and Mott insulating regimes for a small system size (N = 8 particles, Figure 13). We interpreted the observed strong n dependence in the Mott phase in terms of the interaction between holes and doublons. To further explore this physics, we have performed additional simulations for a much larger system (N = 64) far outside the reach of exact diagonalization, and included a new Figure 14.

We now better highlight the physical focus and new results in the substantially modified introduction.

Focusing on the SciPost Physics Acceptance Criteria, we believe our paper "Open(s) a new pathway in an existing or a new research direction, with clear potential for multipronged follow-up work."

While the SWAP estimator had been previously implemented in other quantum Monte Carlo flavors, this has only included those with localized and distinguishable degrees of freedom, e.g. spin models or in the spatial continuum. The extension to lattice models of itinerant bosons that are clearly the most relevant for experimental studies of entanglement (see e.g. Refs. [17-18]) had yet to be accomplished. As evidenced by the lengthy derivation of the SWAP in terms of weights in the continuous imaginary time quantum Monte Carlo algorithm presented here, the generalization to this case is non-trivial. Unlike a spin system, the number of degrees of freedom (worldlines) in a region defined by some subset of spatial modes is a dynamical quantity. As mentioned above, while entanglement in the Bose-Hubbard model has been previously studied in DMRG, these simulations were limited to 1D systems and always implemented a local occupation number restriction (sometimes of only a few bosons per site), resulting in potentially extremely larger errors (of order 100%) in the weakly interacting regime (see e.g. https://journals.aps.org/prb/abstract/10.1103/PhysRevB.105.L121116).

Another benefit of the ground state QMC method presented here is its ability to scale to systems in general dimensions, and based on the comment of referee 1, we have more closely analyzed the sub-leading logarithmic correction to the area law in the 2D Bose-Hubbard model in a 32x32 system composed of 1024 bosons at unit filling. It is predicted that logarithmic corrections should arise due to the existence of a spontaneously broken continuous symmetry in the thermodynamic limit, contributing a universal coefficient due to the presence of a low-energy "tower of states" spectrum and a Goldstone boson. The universal prefactor of this term is known to be related to the number of Goldstone modes, and we report that the value we extract from the QMC is consistent with the value of one for the Bose-Hubbard model. This represents an important first step in studying these, and higher order corrections to the area law which can encode information on the central charges that characterize higher-dimensional conformal field theory, believed to be the fundamental constants that quantify how entropy monotonically decreases under renormalization group flows.

In addition, the growing recent interest in the symmetry resolved entanglement (the number of papers with this in the title/abstract has doubled every year for the last few years) is a strongly motivating factor for our work. The vast majority of these studies have been limited to non-interacting models, as there was simply no technology or method available beyond exact diagonalization for its measurement in interacting systems until now. Computing this quantity, which can have a very large signal to noise ratio in quantum Monte Carlo is no small task, and utilizing the presented algorithm we believe that there is clear potential for follow up work. This may be especially important given the tendency of this quantity to signal quantum phase transitions and its experimental measurement through post-selection. It can also display highly non-trivial dependence on the number of particles in the subsystem (n) and we accentuate this with a new figure 14 in a system of N = 64 bosons.

For these reasons, we believe our substantially modified and refocused manuscript now better highlights our novel physics contribution and thus goes beyond the scope of SciPost Physics Codebases. With the specific reply to the referees we have provided, in combination with the updates to the manuscript, we plan to resubmit our manuscript to SciPost Physics and hope you will continue its consideration for that venue.

---

## Round 3 · List of Changes

• Modified title and abstract to better reflect physics goals of paper
  • Rewrote introduction to include a more physics-based introduction and included a list of specific physics-based novel contributions
  • When introducing the fits of Eqs. (41) and (54), which depend exponentially on the projection length, we now clarify the origin of the exponential behavior.
  • Added 3d diagram illustrating the two-replica configuration space in which we measure second Rényi entropies.
  • For consistency, we now report a rescaled projection length β → βt when reporting results in terms of the dimensionless interaction strength U/t.
  • Numerical values of the fit parameters a,b,c are now reported. The universality of the logarithmic correction is now mentioned and its direct proportionality to the number of Goldstone modes. Relevant literature has been cited.
  • We have added citations to large number of references of these one-dimensional critical point estimations. We also now comment on how a finite size scaling analysis of our S_2 vs. U/t results for large systems could be used to identify the critical point.
  • We now mention that error bars are computed using the jackknife method in the main text.
  • The terminal output is now thoroughly explained in Appendix A and the code's documentation.
  • We have now included an MIT open source license to the code repositories.
  • The code now compiles with g++-12. We have fixed all other issues that were causing compiler warnings. After testing with clang++ and g++-12, no warnings should be given now.
  • All indicated typos have been fixed.

---

## Editorial Decision

published